# DATA INSTANCE PRIOR FOR TRANSFER LEARNING IN GANS

## ABSTRACT

Recent advances in generative adversarial networks (GANs) have shown remarkable progress in generating high-quality images. However, this gain in performance depends on the availability of a large amount of training data. In limited data regimes, training typically diverges, and therefore the generated samples are of low quality and lack diversity. Previous works have addressed training in low data setting by leveraging transfer learning and data augmentation techniques. We propose a novel transfer learning method for GANs in the limited data domain by leveraging informative data prior derived from self-supervised/supervised pre-trained networks trained on a diverse source domain. We perform experiments on several standard vision datasets using various GAN architectures (BigGAN, SNGAN, StyleGAN2) to demonstrate that the proposed method effectively transfers knowledge to domains with few target images, outperforming existing state-of-the-art techniques in terms of image quality and diversity. We also show the utility of data instance prior in large-scale unconditional image generation and image editing tasks.

## 1 INTRODUCTION

Generative Adversarial Networks (GANs) are at the forefront of modern high-quality image synthesis in recent years (Brock et al., 2018; Karras et al., 2020b; 2019). GANs have also demonstrated excellent performance on many related computer vision tasks such as image manipulation (Zhu et al., 2017; Isola et al., 2017), image editing (Plumerault et al., 2020; Shen et al., 2020; Jahanian et al., 2020) and compression (Tschannen et al., 2018). Despite the success in large-scale image synthesis, GAN training suffers from a number of drawbacks that arise in practice, such as training instability and mode collapse (Goodfellow et al., 2016; Arora et al., 2017). It has been observed that the issue of unstable training can be mitigated to an extent by using conditional GANs. However, this is expected as learning the conditional model for each class is easier than learning the joint distribution. The disadvantages of GAN training have prompted research in several non-adversarial generative models (Hoshen et al., 2019; Bojanowski et al., 2018; Li & Malik, 2018; Kingma & Welling, 2014). These techniques are implicitly designed to overcome the mode collapse problem, however, the quality of generated samples are still not on par with GANs.

Current state-of-the-art deep generative models require a large volume of data and computation resources. The collection of large datasets of images suitable for training - especially labeled data in case of conditional GANs - can easily become a daunting task due to issues such as copyright, image quality and also the training time required to get state-of-the-art image generation performance. To curb these limitations, researchers have recently proposed techniques inspired by transfer learning (Noguchi & Harada, 2019; Wang et al., 2018; Mo et al., 2020) and data augmentation methods (Karras et al., 2020a; Zhao et al., 2020b; Zhang et al., 2019). Advancements in data and computation efficiency for image synthesis can enable its applications in data-deficient fields such as medicine (Yi et al., 2019) where labeled data generation can be difficult to obtain.

Transfer learning is a promising area of research (Oquab et al., 2014; Pan & Yang, 2009) that leverages prior information acquired from large datasets to help in training models on a target dataset under limited data and resource constraints. There has been extensive exploration of transfer learning in classification problems that have shown excellent performance on various downstream data-deficient domains. Similar extensions of reusing pre-trained networks for transfer learning (i.e. fine-tuning a

subset of pre-trained network weights from a data-rich domain) have also been recently employed for image synthesis in GANs (Wang et al., 2018; Noguchi & Harada, 2019; Mo et al., 2020; Wang et al., 2020; Zhao et al., 2020a) in the limited data regime. However, these approaches are still prone to overfitting on the sparse target data, and hence suffer from degraded image quality and diversity.

In this work, we propose a simple yet effective way of transferring prior knowledge in unsupervised image generation given a small sample size ($\sim$ 100-2000) of the target data distribution. Our approach is motivated by the formulation of the IMLE technique (Li & Malik, 2018) that seeks to obtain mode coverage of target data distribution by learning a mapping between latent and target distributions using a maximum likelihood criterion. We instead propose the use of data priors in GANs to match the representation of the generated samples to real modes of data. In contrast to (Li & Malik, 2018), we use the images generated using data priors to find the nearest neighbor match to real modes in the generator's learned distribution. In particular, we show that using an informative *data instance prior* in limited and large-scale unsupervised image generation substantially improves the performance in image synthesis. We show that these data priors can be derived from commonly used computer vision pre-trained networks (Simonyan & Zisserman, 2014; Zhang et al., 2018; Noguchi & Harada, 2019; Hoshen et al., 2019) or self-supervised data representations (Chen et al., 2020) (without any violation of the target setting's requirements, i.e. ensuring that the pre-trained network has not been trained on few-shot classes in the few-shot learning setting, for instance). In case of sparse training data, our approach of using data instance priors leverages a model pre-trained on a rich source domain to the learn the target distribution. Different from previous works (Noguchi & Harada, 2019; Wang et al., 2020; 2018) which rely on fine-tuning models trained on a data-rich domain, we propose to leverage the feature representations of our source model as data instance priors, to distill knowledge (Romero et al., 2015; Hinton et al., 2015) into our target generative model.

We note that our technique of using data instance priors for transfer learning becomes fully unsupervised in case the data priors are extracted from self-supervised pre-trained networks. Furthermore, in addition to image generation in low data domain, we also achieve state-of-the-art Fréchet inception distance (FID) score (Heusel et al., 2017) on large-scale unsupervised image generation and also show how this framework of transfer learning supports several image editing tasks.

We summarize our main contributions as follows:

- We propose Data Instance Prior (DIP), a novel transfer learning technique for GAN image synthesis in low-data regime. We show that employing DIP in conjunction with existing few-shot image generation methods outperforms state-of-the-art results. We show with as little as 100 images our approach DIP results in generation of diverse and high quality images (see Figure 3).
- We demonstrate the utility of our approach in large-scale unsupervised GANs (Miyato et al., 2018; Brock et al., 2018) achieving the new state-of-the-art in terms of image quality (Heusel et al., 2017) and diversity (Sajjadi et al., 2018; Metz et al., 2017).
- We show how our framework of DIP by construction enables inversion of images and common image editing tasks (such as cutmix, in-painting, image translation) in GANs.

We call our method a *data instance prior* (and not just data prior), since it uses representations of instances as a prior, and not a data distribution itself.

## 2 RELATED WORK

**Deep Generative Models** In recent years, there has been a surge in the research of deep generative models. Some of the popular approaches include variational auto-encoders (VAEs) (Rezende et al., 2014; Kingma & Welling, 2014), auto-regressive (AR) models (Van Oord et al., 2016; Van den Oord et al., 2016) and GANs (Goodfellow et al., 2014). VAE models learn by maximizing the variational lower bound of likelihood of generating data from a given distribution. Auto-regressive approaches model the data distribution as a product of the conditional probabilities to sequentially generate data. GANs comprise of two networks, a generator and a discriminator that train in a min-max optimization. Specifically, the generator aims to generate samples to fool the discriminator, while the discriminator learns distinguish these generated samples from the real samples. Several research efforts in GANs have focused around improving the performance (Karras et al., 2018; Denton et al., 2015; Radford et al., 2016; Karras et al., 2020b; 2019; Brock et al., 2018; Zhang et al., 2019) and

training stability (Salimans et al., 2016b; Gulrajani et al., 2017; Arjovsky et al., 2017; Miyato et al., 2018; Mao et al., 2017; Chen et al., 2019). Recently, the areas of latent space manipulation for semantic editing (Shen et al., 2020; Jahanian et al., 2020; Zhu et al., 2020; Plumerault et al., 2020) and few-shot image generation (Wang et al., 2020; Mo et al., 2020; Noguchi & Harada, 2019) have gained traction in an effort to mitigate the practical challenges while deploying GANs. Several other non-adversarial training approaches such as (Hoshen et al., 2019; Bojanowski et al., 2018; Li & Malik, 2018) have also been explored for generative modeling, which leverage supervised learning along with perceptual loss (Zhang et al., 2018) for training such models.

**Transfer Learning in GANs**   While there has been extensive research in the area of transfer learning for classification models (Yosinski et al., 2014; Oquab et al., 2014; Tzeng et al., 2015; Pan & Yang, 2009; Donahue et al., 2014), relatively fewer efforts have explored this on the task of data generation (Wang et al., 2018; 2020; Noguchi & Harada, 2019; Zhao et al., 2020a; Mo et al., 2020). (Wang et al., 2018) proposed to fine-tune a pre-trained GAN model (often having millions of parameters) from a data-rich source to adapt to the target domain with limited samples. This approach, however, often suffers from overfitting as the final model parameters are updated using only few samples of the target domain. To counter overfitting, the work of (Noguchi & Harada, 2019) proposes to update only the batch normalization parameters of the pre-trained GAN model. In this approach, however, the generator is not adversarially trained and uses supervised $L_1$ pixel distance loss and perceptual loss (Johnson et al., 2016; Zhang et al., 2018) which often leads to generation of blurry images in the target domain. Based on the assumption that source and target domain support sets are similar, (Wang et al., 2020) recently proposed to learn an additional mapping network that transforms the latent code suitable for generating images of target domain while keeping the other parameters frozen. We compare against all leading baselines including (Noguchi & Harada, 2019; Wang et al., 2020) on their respective tasks, and show that our method DIP outperforms them substantially, while being simple to implement.

A related line of recent research aims to improve large-scale unsupervised image generation in GANs by employing self-supervision - in particular, an auxiliary task of rotation prediction (Chen et al., 2019) or using one-hot labels obtained by clustering in the discriminator's (Liu et al., 2020) or ImageNet classifier feature space (Sage et al., 2018). In contrast, our method utilizes data instance priors derived from the feature activations of self-supervised/supervised pre-trained networks to improve unsupervised few-shot and large-scale image generation, leading to simpler formulation and higher performance as shown in our experiments in Section 5.3 and Appendix A. Recently, some methods (Karras et al., 2020a; Zhao et al., 2020b; Zhang et al., 2019; Zhao et al., 2020c) have leveraged data augmentation to effectively increase the number of samples and prevent overfitting in GAN training. However, data augmentation techniques often times alter the true data distribution and there is a leakage of these augmentations to the generated image, as shown in (Zhao et al., 2020c;b). To overcome this, (Zhao et al., 2020b) recently proposed to use differential augmentation and (Karras et al., 2020a) leveraged an adaptive discriminator augmentation mechanism. We instead focus on leveraging informative data instance priors, and in fact, show how our DIP method can be used in conjunction with (Zhao et al., 2020b) to improve performance.

## 3   PRELIMINARIES

We now present the related background on generative models that are essential for our methodology.

**Conditional GAN:**   It consists of a generator $G : \mathbb{R}^m \times Y \to \mathbb{R}^p$ and a discriminator $D : \mathbb{R}^p \times Y \to [0, 1]$ which are trained adversarially to learn a target data distribution $q(\mathbf{x}|y)$, where $\mathbf{x} \in \mathbb{R}^p$, $y \in Y$, the space of class labels. The goal of $G$ is to generate samples from noise $\mathbf{z} \sim p(\mathbf{z})$, $\mathbf{z} \in \mathbb{R}^m$ given a conditional label $y \sim q(y)$ that matches the target distribution and the aim of $D$ is to distinguish between real samples and those generated from $G$. Conditional GANs use class-level information $y$ of data in the generator and discriminator. The standard hinge loss (Tran et al., 2017) for training GANs is given by:

$$L_D = \mathbb{E}_{y \sim q(y)} \big[ \mathbb{E}_{\mathbf{x} \sim q(\mathbf{x}|y)} [\max(0, 1 - D(\mathbf{x}, y))] \big] + \mathbb{E}_{y \sim q(y)} \big[ \mathbb{E}_{\mathbf{z} \sim p(\mathbf{z})} [\max(0, 1 + D(G(\mathbf{z}|y), y))] \big]$$
$$L_G = -\mathbb{E}_{y \sim q(y)} \big[ \mathbb{E}_{\mathbf{z} \sim p(\mathbf{z})} [D(G(\mathbf{z}|y), y)] \big]$$

(1)

where the discriminator score $D(\mathbf{x}, y)$ depends on input image (either real or fake) and its class $y$ (Miyato & Koyama, 2018; Odena et al., 2017). Generally, the class information is passed into $G$ through a one-hot vector concatenated with $z$ or through conditional batch norm layers (De Vries et al., 2017; Dumoulin et al., 2016).

**Implicit Maximum Likelihood Estimation**  IMLE (Li & Malik, 2018) is a non-adversarial technique that uses a maximum likelihood criterion to train the generative model. During training, each real sample of the data distribution is matched to a generated image through nearest neighbour search and the distance between the two is minimized. The optimization objective to update the parameters of network $G$ in each training step is given by:

$$\min \mathbb{E}_{\mathbf{z_1}...\mathbf{z_m} \sim \mathcal{N}(\mathbf{0}, \mathbf{I}_d)} \left[ \mathbb{E}_{\mathbf{x} \sim q(\mathbf{x})} \left[ \|G(\mathbf{z}^*(\mathbf{x})) - \mathbf{x}\|_2^2 \right] \right]$$
$$\text{where } \mathbf{z}^* = \min_{\mathbf{z_1}...\mathbf{z_m}} \|G(\mathbf{z}) - \mathbf{x}\|_2^2 \tag{2}$$

Here, the inner optimization of finding latent code $\mathbf{z}^*(\mathbf{x})$ from the batch $\{\mathbf{z_1}...\mathbf{z_m}\}$ such that $G(\mathbf{z}^*(\mathbf{x}))$ is nearest to the real image $\mathbf{x}$ from the data distribution is implemented using the nearest neighbor search technique (Li & Malik, 2017). The above objective promotes that each real example is close to some generated sample from the trained generator.

## 4  DIP: PROPOSED METHODOLOGY

We propose a transfer learning framework DIP for training GANs that exploits knowledge extracted from self-supervised/supervised networks, pre-trained on a rich and diverse source domain in the form of data instance priors. It has been shown that providing class label information in GANs significantly improves training stability and quality of generated images as compared to unconditional setting (Miyato & Koyama, 2018; Chen et al., 2019). However, in practice, GANs are observed to be prone to mode-collapse that is further exacerbated in case of sparse training data. We take motivation from the reconstructive framework of IMLE (Li & Malik, 2018) and propose to condition GANs on image instance priors that act as a regularizer to prevent mode collapse and discriminator overfitting.

**Knowledge Transfer in GAN**  GANs are a class of implicit generative models that minimize a divergence measure between the data distribution $q(\mathbf{x})$ and the generator output distribution $G(\mathbf{z})$ where $\mathbf{z} \sim p(\mathbf{z})$ denotes the latent distribution. Intuitively, this minimization of a divergence objective ensures that each generated sample $G(\mathbf{z})$ is close to some data example $\mathbf{x} \sim q(\mathbf{x})$. However, this does not ensure the converse, i.e. that each data instance is close to a generated sample, which can result in mode dropping. To counter this, especially in limited data regime, we propose to update the parameters of the model so that each real data example is close to some generated sample similar to (Li & Malik, 2018) by using data instance priors as conditional label in GANs. Moreover to enable transfer of knowledge, image features extracted from networks pre-trained on a large source domain are used as the instance level prior.

Given a pre-trained feature extractor $C : \mathbb{R}^p \to \mathbb{R}^d$, $\mathbf{x} \in \mathbb{R}^p$, which is trained on a source domain using supervisory signals or self-supervision, we use its output $C(\mathbf{x})$ as condition in the generator. $G$ is conditioned on $C(\mathbf{x})$ using conditional batch-norm (Dumoulin et al., 2016) whose input is $G_{emb}(C(\mathbf{x}))$, where $G_{emb}$ is a learnable matrix. During training we enforce that $G(\mathbf{z}|C(\mathbf{x}))$ is close to the real image $\mathbf{x}$ in discriminator feature space (similar to $G(\mathbf{z}^*(\mathbf{x}))$ being close to $x$ in Eq 2). Let the discriminator be $D = D_l \circ D_f$ ($\circ$ denotes composition) where $D_f$ is discriminator's last feature layer and $D_l$ is the final linear layer. To enforce the above objective we map $C(\mathbf{x})$ to the dimension equal to discriminator's feature layer $D_f$ using a trainable matrix $D_{emb}$ and minimize distance between $D_{emb}(C(\mathbf{x}))$ and $D_f$ of both real image $\mathbf{x}$ and generated image $G(\mathbf{z}|C(\mathbf{x}))$ in an adversarial manner. Hence, our final GAN training loss for the discriminator and generator is given by:

$$L_D = \mathbb{E}_{\mathbf{x} \sim q(x)}[\max(0, 1 - D(\mathbf{x}, C(\mathbf{x})))] + \mathbb{E}_{\mathbf{x} \sim q(x), \mathbf{z} \sim p(\mathbf{z})}[\max(0, 1 + D(G(\mathbf{z}|C(\mathbf{x})), C(\mathbf{x})))]$$
$$L_G = -\mathbb{E}_{\mathbf{x} \sim q(x), \mathbf{z} \sim p(\mathbf{z})}[D(G(\mathbf{z}|C(\mathbf{x})), C(\mathbf{x}))]$$
$$\tag{3}$$

where

$$D(\mathbf{x}, \mathbf{y})) = D_{emb}(\mathbf{y}) \cdot D_f(\mathbf{x}) + D_l \circ D_f(\mathbf{x}) \tag{4}$$

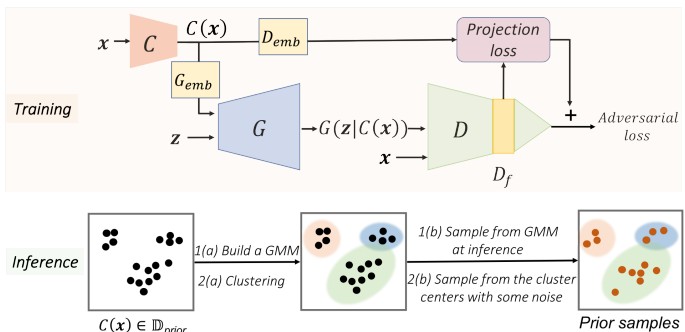

Figure 1: Overview of our proposed technique, Data Instance Priors (DIP) for transfer learning in GANs. *Top:* DIP training with feature $C(\mathbf{x})$ of a real sample $\mathbf{x}$ as a conditional prior in the conditional GAN framework of (Miyato & Koyama, 2018). $C$ is a pre-trained network on a rich source domain from which we wish to transfer knowledge. *Bottom*: Inference over trained GAN involves learning a distribution over the set of training data prior $\{C(\mathbf{x})\}$ to enable sampling of conditional priors.

---

**Algorithm 1:** Data Instance Prior Training (DIP)

---

1 **Input**:$G$, $D$ network with parameters $\theta_G$ and $\theta_D$, pre-trained model $C$ for extracting prior condition, samples from real data distribution $q(x)$ and latent distribution $p(z)$, batch size $b$, number of training iterations, discriminator update steps $d_{step}$ for each generator update, Adam hyperparameters $\alpha, \beta_1, \beta_2$.

2 **for** *number of training iterations* **do**

3      **for** $t : 1...d_{step}$ **do**

4          Sample batch $x \sim q(x)$, $z \sim p(z)$

5          $x_{fake} = G(z|C(x))$

6          Calculate $D(x, C(x)) = D_f(x) \cdot D_{emb}(C(x)) + D_l \circ D_f(x)$

7          Calculate $D(x_{fake}, C(x)) = D_f(x_{fake}) \cdot D_{emb}(C(x)) + D_l \circ D_f(x_{fake})$

8          $L_D = \max(0, 1 - D(x, C(x))) + \max(0, 1 + D(x_{fake}, C(x)))$

9          Update $\theta_D \leftarrow Adam(L_D, \alpha, \beta_1, \beta_2)$

10      **end**

11      Sample $z \sim p(z)$

12      Generate images $x_{fake} = G(z|C(x)$

13      Calculate $D(x_{fake}, C(x)) = D_f(x_{fake}) \cdot D_{emb}(C(x)) + D_l \circ D_f(x_{fake})$

14      $L_G = -D(x_{fake}, C(x))$

15      Update $\theta_G \leftarrow Adam(L_G, \alpha, \beta_1, \beta_2)$

16 **end**

17 **return** $\theta_G, \theta_D$.

---

In the above formulation, the first term in Eq. 4 is the projection loss as in (Miyato & Koyama, 2018) between input image and conditional embedding of discriminator. Since conditional embedding is extracted from a pre-trained network, above training objective leads to feature level knowledge distillation from $C$. It also acts as a regularizer on the discriminator reducing its overfitting in the limited data setting as shown in Figure 2. The gap between discriminator score ($D_l \circ D_f$) of training and validation images keeps on increasing and FID quickly degrades for baseline model as compared to DIP when trained on only 10% data of CIFAR-100. Moreover, enforcing feature $D_f(G(\mathbf{z}|C(\mathbf{x})))$ to be similar to $D_{emb}(C(\mathbf{x}))$ promotes that for each real sample, there exists a generated sample close to it and hence promotes mode coverage of target data distribution. We demonstrate that the above proposed use of data instance priors from a pre-trained feature extractor, while designed for a limited data setting, also benefits in large-scale image generation. Our overall methodology is illustrated in Figure 1 and pseudo code is in Algorithm 1.

**Random image generation at inference**    Given the training set $\mathbb{D}_{image} = \{\mathbf{x}_j\}_{j=1}^n$ of sample size $n$ and its corresponding training data prior set $\mathbb{D}_{prior} = \{C(\mathbf{x}_j)\}_{j=1}^n$, the generator requires access to $\mathbb{D}_{prior}$ for sample generation. In case of few-shot image generation where size of $\mathbb{D}_{prior}$ is limited ($\sim 100$), to create more variations we generate images conditioned on interpolation of two randomly sampled prior i.e.

$$G(\mathbf{z}|\lambda \cdot \mathbf{p_1} + (1 - \lambda) \cdot \mathbf{p_2}) \text{ where } \mathbf{p_1}, \mathbf{p_2} \sim \mathbb{D}_{prior}; \ \ \lambda \sim Uniform[0, 1] \qquad (5)$$

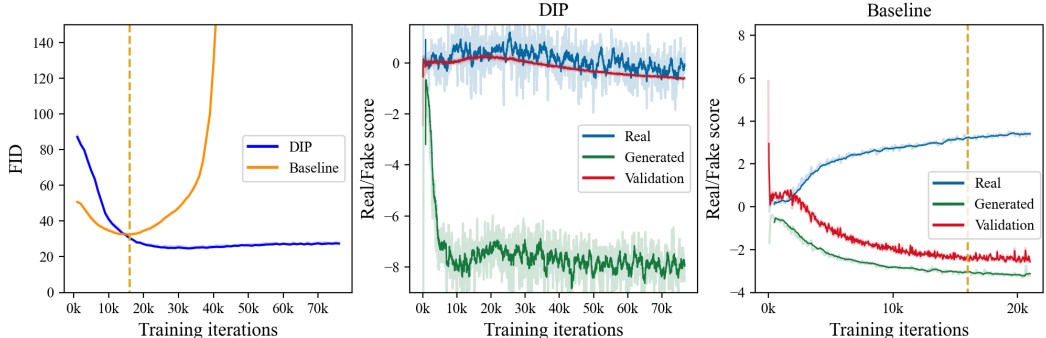

Figure 2: Comparison between DIP and Baseline when trained on 10% data of CIFAR-100. *left*: FID (in Pytorch) of baseline training starts increasing very early in training (around 15k) as compared to FID of DIP training. *middle*: Discriminator score on training and validation images remain similar to each other and consistently higher than score of generated images for DIP model. *right*: Discriminator score on training and validation images start diverging and training collapses for the baseline model.

In case of large-scale image generation, to avoid storing $\mathbb{D}_{prior}$ corresponding to complete training set (possibly in order of millions), we propose to cluster (Sculley, 2010) or build a Gaussian Mixture Model (GMM) (Xu & Jordan, 1996) on $\mathbb{D}_{prior}$ and store only the cluster centroids and thus enable memory efficient sampling of conditional prior from the distribution fit during inference as follows:

$$G(\mathbf{z}|\mu + \mathcal{N}(\mathbf{0}, \mathbf{I}_k)) \text{ where } \mu \sim \text{K-MeansCentroids}(G_{emb}(\mathbb{D}_{prior}))$$
$$\text{or } G(\mathbf{z}|\mathcal{N}(\mu, \mathbf{\Sigma})) \text{ where } \mu, \mathbf{\Sigma} \sim \text{GMM}(G_{emb}(\mathbb{D}_{prior})) \tag{6}$$

**Controlled image generation through semantic diffusion** We observed that high-level semantics (e.g. smile, hair, gender, glasses, etc in case of faces) of a generated image, $G(\mathbf{z}|C(\mathbf{x}))$, relied on the conditional prior, $C(\mathbf{x})$. Complementarily, variations in the latent code $\mathbf{z} \sim \mathbf{N}(0, I)$ induced fine-grained changes such as skin texture, face shape, etc. This suggests that one can exploit our conditional prior, $C(\mathbf{x})$, to get some control over the image generation's high-level semantics. We show that by altering an image $\mathbf{x}$ (through CutMix, CutOut, etc) and using $C(\mathbf{x})$ of the altered image as our new input prior helps in generating samples with the desired attributes, as shown in Fig 5. In a similar manner, DIP also allows generation of images with certain cues (like sketch to image generation, as shown in Fig 5 and Appendix). We note that the generation of samples at inference, in this case, can simply be done by using $C(\mathbf{x})$ as condition in $G$.

## 5 EXPERIMENTS

We perform extensive experiments to highlight the efficacy of our data instance prior module DIP in unsupervised training based on SNGAN (Miyato et al., 2018), Big-GAN (Brock et al., 2018) and StyleGAN2 (Karras et al., 2020b) architectures. For extracting image prior information, we use the last layer activations of: (1) Vgg16 (Simonyan & Zisserman, 2014) classification network trained on ImageNet; and (2) SimCLR (Chen et al., 2020) network trained using self-supervision on ImageNet. We conduct experiments on few-shot ($\sim$ 25-100 images), limited ($\sim$ 2k-5k images) and large-scale ($\sim$ 50k-1M images) data settings. For evaluation, we use FID (Heusel et al., 2017), precision and recall scores (Kynkäänniemi et al., 2019) to assess the quality and mode-coverage/diversity of the generated images.

### 5.1 FEW-SHOT IMAGE GENERATION

**Baselines and Datasets** For comprehensive evaluation, we compare and augment our methodology DIP with training SN-GAN from scratch and the following leading baselines: Batch Statistics Adaptation (BSA) (Noguchi & Harada, 2019), TransferGAN (Wang et al., 2018), FreezeD (Mo et al., 2020) and DiffAugment (Zhao et al., 2020b). In case of BSA, a non-adversarial variant, GLANN (Hoshen et al., 2019) is used which optimizes for image embeddings and generative model

| | | SN-GAN (128 x 128) | | | | | | | | |
| Method | Pre-training | Anime | | | Faces | | | Flower | | |
| | | FID ↓ | P ↑ | R ↑ | FID ↓ | P ↑ | R ↑ | FID ↓ | P ↑ | R ↑ |
| From scratch | ✗ | 120.38 | 0.61 | 0.00 | 140.66 | 0.31 | 0.00 | 124.02 | 0.30 | 0.09 |
| + DIP-Vgg16 | | **66.85** | **0.71** | **0.03** | **68.49** | **0.74** | **0.15** | **94.22** | **0.62** | **0.41** |
| TransferGAN | ✓ | 102.75 | **0.70** | 0.00 | 101.15 | **0.85** | 0.00 | 113.35 | 0.71 | 0.09 |
| + DIP-Vgg16 | | **86.96** | 0.57 | **0.02** | **75.21** | 0.70 | **0.10** | **110.24** | 0.55 | **0.11** |
| FreezeD | ✓ | 109.40 | **0.67** | 0.00 | 107.83 | **0.83** | 0.00 | **91.80** | **0.69** | 0.14 |
| + DIP-Vgg16 | | 93.36 | 0.56 | **0.03** | 77.09 | 0.68 | 0.14 | 120.43 | 0.53 | 0.20 |
| + DIP-SimCLR | | **89.39** | 0.46 | 0.025 | **70.40** | 0.74 | **0.22** | 120.13 | 0.63 | **0.33** |
| DiffAugment | ✗ | 85.16 | **0.95** | 0.00 | 109.25 | **0.84** | 0.00 | 83.45 | 0.75 | 0.23 |
| + DIP-Vgg16 | | **48.67** | 0.82 | 0.03 | **62.44** | 0.80 | 0.19 | **79.86** | **0.79** | **0.57** |
| + DIP-SimCLR | | 52.41 | 0.77 | **0.04** | 64.53 | 0.78 | **0.22** | 87.92 | 0.75 | 0.54 |
| BSA* | ✓ | 92.0 | - | - | 123.2 | - | - | 129.8 | - | - |
| GLANN+DIP-Vgg16 | | **67.07** | **0.87** | **0.01** | **60.11** | **0.95** | **0.08** | **97.73** | **0.95** | **0.51** |

Table 1: Few-shot image generation results using 100 training images (↓: lower is better; ↑: higher is better). Precision and Recall scores are based on (Kynkäänniemi et al., 2019). FID is computed between 10k, 7k, 5k generated and 10k, 7k, 251 real samples for Anime, Faces and Flower respectively. * denotes directly reported from the paper.

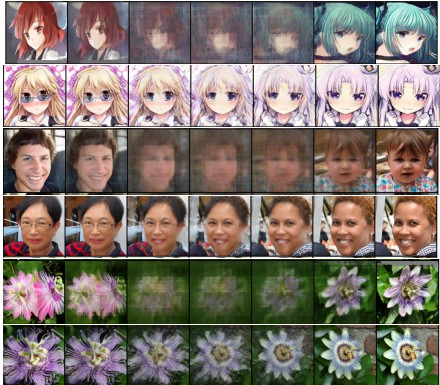

Figure 3: 100-shot image interpolation between instance-level prior for DiffAugment (Zhao et al., 2020b) *(Rows 1,3,5)* and DiffAugment + DIP-Vgg16 *(Rows 2,4,6)* on Anime, Faces and Flower datasets respectively.

| | Big-GAN (128 x 128) | | |
| Method | Places2.5k | FFHQ2k | CUB6k |
| | FID ↓ | FID ↓ | FID ↓ |
| MineGAN | 75.50 | 75.91 | 69.64 |
| TransferGAN | 162.91 | 126.23 | 138.87 |
| + DIP-Vgg16 | **57.35** | **44.43** | **23.37** |
| FreezeD | 191.04 | 161.87 | 142.47 |
| + DIP-Vgg16 | **50.58** | **43.90** | **26.90** |
| DiffAugment | 56.48 | 31.60 | 36.09 |
| + DIP-Vgg16 | 30.76 | 23.19 | 15.81 |
| + DIP-SimCLR | **26.65** | **21.06** | **12.36** |

Table 2: Limited data image generation using different approaches. FID (lower is better) is computed between 10k, 7k, 6k generated and real samples (disjoint from training set) for Places2.5k, FFHQ2k, CUB datasets respectively. BigGAN pre-trained on ImageNet is fine-tuned in all approaches.

through perceptual loss[1]. We use our data priors to distill knowledge over these image embeddings. For more training and hyperparameter details, please refer to Appendix A.

We perform experiments on randomly chosen 100 images at $128 \times 128$ resolution from: (1) Anime[2] (2) FFHQ (Karras et al., 2019) and (3) Oxford 102 flowers (Nilsback & Zisserman, 2008) (we restrict to the Passion flower class following (Noguchi & Harada, 2019), to avoid overlap with ImageNet classes) datasets. The above datasets are chosen to ensure that there is no class label intersection with ImageNet classes. For methods with pre-training, we finetune SNGAN pre-trained on ImageNet as done in (Noguchi & Harada, 2019). We also show additional results at $256 \times 256$ resolution as well as additional datasets (Pandas, Grumpy Cat, Obama) with StyleGAN-2 (Karras et al., 2020b) in Appendix A.

**Results** Using DIP shows consistent improvement in FID and Recall over all baseline methods as shown in Table 1. Fig 3 shows samples generated via interpolation between conditional embedding of models trained via DIP-Vgg on DiffAugment and vanilla DiffAugment. These results show qualitatively the significant improvement obtained using our DIP-based transfer learning approach. Comparatively, the baseline, vanilla DiffAugment, fails to generate realistic interpolation and for

---

[1] The code provided with BSA was not reproducible, and hence this choice
[2] www.gwern.net/Danbooru2018

| Method | Inference | CIFAR-10 | | | CIFAR-100 | | | FFHQ | | | LSUN-Bedroom | | | ImageNet32x32 | | |
|---|---|---|---|---|---|---|---|---|---|---|---|---|---|---|---|---|
| | | FID ↓ | P ↑ | R ↑ | FID ↓ | P ↑ | R ↑ | FID ↓ | P ↑ | R ↑ | FID ↓ | P ↑ | R ↑ | FID ↓ | P ↑ | R ↑ |
| Baseline | | 19.73 | 0.64 | 0.70 | 24.66 | 0.61 | 0.67 | 21.67 | 0.77 | 0.47 | 9.89 | 0.58 | 0.42 | 16.19 | 0.60 | **0.67** |
| SSGAN | | 15.65 | 0.67 | 0.68 | 21.02 | 0.61 | 0.65 | - | - | - | 7.68 | 0.59 | 0.50 | 17.18 | 0.61 | 0.65 |
| Self-Cond GAN | | 16.72 | 0.71 | 0.64 | 21.8 | 0.64 | 0.60 | - | - | - | - | - | - | 15.56 | 0.66 | 0.63 |
| | $\mathbb{D}_{prior}$ | 10.57 | 0.75 | 0.65 | 14.11 | 0.70 | 0.66 | 13.88 | 0.85 | 0.55 | 3.77 | 0.67 | 0.57 | 9.93 | 0.66 | 0.63 |
| DIP | GMM | **11.24** | **0.74** | 0.64 | **15.71** | **0.70** | 0.62 | 15.83 | 0.76 | **0.55** | 4.99 | 0.66 | **0.54** | 12.11 | **0.64** | 0.62 |
| Vgg16 | K-means | 11.79 | 0.70 | 0.69 | 15.75 | 0.67 | 0.69 | **13.43** | 0.82 | 0.54 | **4.69** | 0.71 | 0.50 | **11.96** | 0.62 | 0.64 |
| | $\mathbb{D}_{prior}$ | 12.60 | 0.69 | 0.70 | 16.26 | 0.67 | 0.70 | 13.50 | 0.83 | 0.59 | 3.85 | 0.67 | 0.56 | 11.43 | 0.62 | 0.68 |
| DIP | GMM | 14.42 | 0.68 | 0.65 | 20.08 | 0.67 | 0.62 | 16.62 | 0.77 | 0.53 | 4.92 | 0.62 | 0.53 | 14.99 | 0.60 | 0.63 |
| SimCLR | K-means | 14.27 | 0.67 | **0.70** | 17.96 | 0.66 | **0.699** | 14.10 | 0.82 | 0.52 | 5.48 | **0.74** | 0.45 | 15.58 | 0.62 | 0.64 |

Table 3: Comparison of FID, Precision and Recall metrics of DIP with Baseline and SSGAN for large-scale image generation. Best values obtained by using complete training set $D_{prior}$ are underlined and the best value among all other approaches are in bold.

the most part, presents memorized training set images. DIP performs better when training is done from scratch as compared to FreezeD and TransferGAN but is worse than DiffAugment+DIP. We present additional ablation studies in Appendix A, including more qualitative comparisons, to study the benefit of using DIP in few-shot image generation.

## 5.2 LIMITED DATA IMAGE GENERATION

In many practical scenarios, we have access to moderate number of images (1k-5k) instead of just a few examples, however the limited data may still not be enough to achieve stable GAN training. We show the benefit of our approach in this setting and compare our results with: MineGAN(Wang et al., 2020), TransferGAN, FreezeD, and DiffAugment. We perform experiments on three $128 \times 128$ resolution datasets: FFHQ2k, Places2.5k and CUB6k following (Wang et al., 2020). FFHQ2k contains 2K training samples from FFHQ (Karras et al., 2019) dataset. Places2.5k is a subset of Places365 dataset (Zhou et al., 2014) with 500 examples each sampled from 5 classes (alley, arch, art gallery, auditorium, ball-room). CUB6k is the complete training split of CUB-200 dataset (Wah et al., 2011). We use the widely used BigGAN (Brock et al., 2018) architecture, pre-trained on ImageNet for fine-tuning. Table 2 shows our results; using DIP consistently improves FID on existing baselines by a significant margin. More implementation details are given in Appendix B and sample generated images via our approach are shown in Fig 8. We also compare our approach with DiffAugment on CIFAR-10 and CIFAR-100 dataset while varying the number of sample used for training in Table 9 Appendix B.

## 5.3 LARGE-SCALE IMAGE GENERATION

In order to show the usefulness of our method on large-scale image generation, we carry out experiments on CIFAR-10, CIFAR-100 (Krizhevsky et al., 2010) and ImageNet-$32 \times 32$ datasets with 50k, 50k and $\sim 1.2M$ training images respectively at $32 \times 32$ resolution. For a higher $128 \times 128$ resolution, we perform experiments on FFHQ and LSUN-bedroom (Yu et al., 2015) datasets with 63k and 3M training samples. We use a ResNet-based architecture for both discriminator and generator similar to BigGAN (Brock et al., 2018) for all our experiments. We also compare DIP with SS-GAN (Chen et al., 2019) and Self-Conditional GAN (Liu et al., 2020). Implementation and training hyperparameter details are provided in Appendix C.

Table 3 reports the FID, precision and recall score on the generated samples and the test set for baselines and our approach (DIP). For K-means clustering on $G_{emb}(\mathbb{D}_{prior})$, we set number of clusters to $10K$ and for GMM, the number of components are fixed to $1K$ for all datasets. DIP achieves better FID, precision and recall scores compared to leading baselines. Sample qualitative results are shown in the Appendix (Figure 11). To further analyze the role of prior in our methodology, we train CIFAR-100 dataset with DIP using priors from different pre-trained models. As shown in the results in Table 5, the FID metric remains relatively similar for different priors when compared to the baseline. We also evaluate the quality of inverted images for $128 \times 128$ resolution on FFHQ and LSUN datasets using Inference via Optimization Measure (IvOM) (Metz et al., 2017) to emphasize the high instance-level data coverage in the prior space of GANs trained through our approach (details on IvOM calculation are provided in Appendix C). Table 4 shows the IvOM and FID metric between inverted and real query images. Figure 4 shows sample inverted images. We observe both

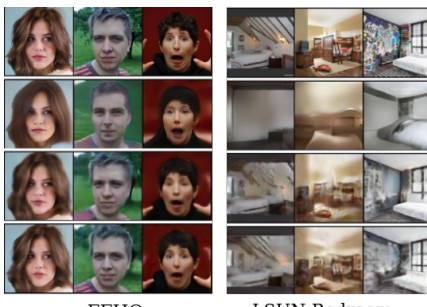

FFHQ      LSUN-Bedroom

Figure 4: Images generated through IvOM for randomly sampled test set images on FFHQ and LSUN-Bedroom. *(Top to Bottom:)* Original images, Baseline, Baseline + DIP-Vgg16, Baseline + DIP-SimCLR.

| Method | FFHQ | | LSUN-Bedroom | |
|---|---|---|---|---|
| | IvOM ↓ | FID ↓ | IvOM ↓ | FID ↓ |
| Baseline | 0.0386 | 85.06 | 0.0517 | **115.02** |
| + DIP-Vgg16 | 0.0142 | 73.85 | 0.0191 | 129.4 |
| + DIP-SimCLR | **0.0125** | **71.44** | **0.0161** | 116.11 |

Table 4: IvOM and FID measure on 500 random test images of FFHQ and LSUN-Bedroom datasets

| Method | CIFAR-100 |
|---|---|
| Baseline | 24.66 |
| + DIP-SimCLR (ImageNet) | 16.26 |
| + DIP-SimCLR (CIFAR-100) | 14.62 |
| + DIP-ResNet50 (Places-365) | 14.68 |

Table 5: Comparison of FID when using prior from various pre-trained models on CIFAR-100

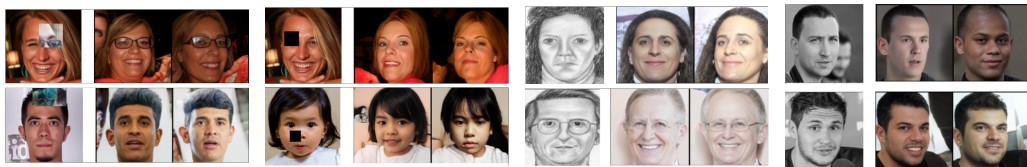

Figure 5: Semantic diffusion for image manipulation using DIP-Vgg16 model on FFHQ dataset. *(Left to Right:)* Custom Editing, Inpainting, Sketch-to-Image Translation and Colorization.

from qualitative and quantitative perspective, models trained via DIP invert a given query image better than the corresponding baselines.

**Semantic Diffusion and Variations** As described in Section 4, our conditional DIP module provides us with a framework to alter input images and thus generate images with specific semantics. Figure 5 demonstrates how controlled semantic diffusion can be leveraged in several image manipulation tasks. We perform: (1) *Custom Editing* by using CutMix (i.e pasting a desired portion of one image upon another); (2) *Inpainting* by using CutOut (i.e removing random portions in example image); (3) *Sketch-to-Image* by providing the feature of a sketch as conditional prior; and (4) *Colorization* by using the feature of a given grayscale image for conditioning. As evident from Figure 5, our trained generator is able to generalize well through its ability to diffuse the semantic information from altered (cutmix and cutout) as well as out-of-domain (sketches and gray-scale) images. For more qualitative results, please see Fig 10 in Appendix. We can also use interpolation, noise injection and Mixup in the conditional space to generate meaningful variations of a given image as shown in Fig 9 in the Appendix.

## 6   CONCLUSION

In this work, we present a novel data instance prior based transfer learning approach to improve the quality and diversity of images generated using GANs when a few training data samples are available. By leveraging features as priors from rich source domain in limited unsupervised image synthesis, we show the utility of our simple yet effective approach on various standard vision datasets and GAN architectures. We demonstrate the efficacy of our approach in image generation with limited data, where it achieves the new state-of-the performance, as well as on large-scale settings. Furthermore, using our framework of training via instance level priors, we show how easily we can perform common image editing tasks by manipulating these priors. As future work, it would be interesting to explore the application of prior information in other potential image editing tasks.

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

# Appendix

## A FEW-SHOT IMAGE GENERATION

**Performance on varying number of training images**   We vary the number of training examples in Anime dataset from 25-500 for baseline few-shot algorithms and their respective augmentations with DIP-Vgg16. The FID metric comparison in Fig 6a shows the benefit of our approach when used with existing training algorithms. The FID metric for all approaches improves (decreases) with the increase the number of training images with DIP out-performing corresponding baselines. Sample images generated by our approach are shown in Fig 6b.

**Memorization in GANs**   To evaluate whether trained GANs are actually generating novel images instead of only memorizing the training set, we calculate FID between images randomly sampled from training set with repetition and the separate test set for Anime and FFHQ dataset. For Anime dataset, we get an FID of $81.23$ and for FFHQ, $100.07$. On comparing these numbers to Table 1 we observe that only on using DIP with existing algorithms, we achieve a better FID score suggesting the usefulness of the proposed approach.

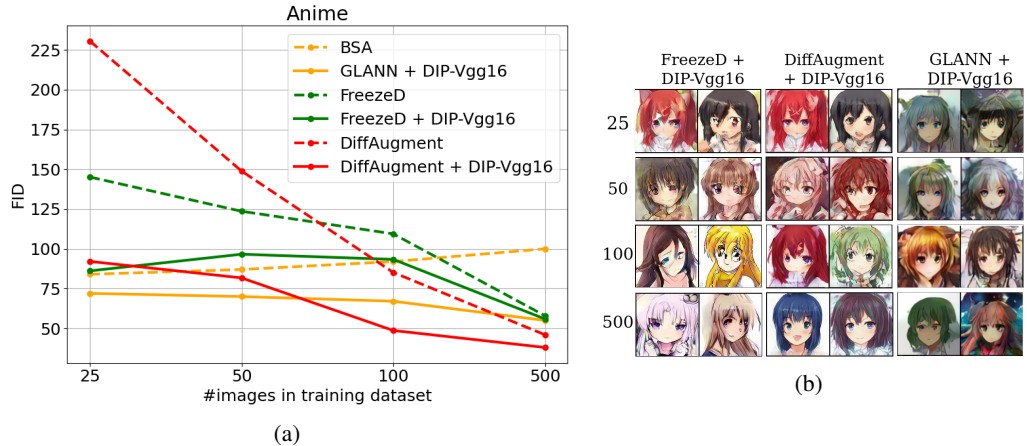

(a)

(b)

Figure 6: (a) FID (lower is better) performance graph of few-shot image generation on 25-500 images of Anime dataset using various approaches on SNGAN model; (b) Samples of few-shot image generation on 25-500 images of Anime dataset using DIP on various approaches on SNGAN.

**Few-shot image generation with StyleGAN-2**   For 256 x 256 resolution dataset with StyleGAN-2 architecture, we follow (Zhao et al., 2020b) and perform experiments on 100-shot Obama, Panda and Grumpy Cat dataset with pre-trained models on FFHQ (Karras et al., 2019) dataset. Table 6 shows consistent improvement in FID scores when trained with DIP irrespective of baseline training algorithm except on Grumpy Cat dataset. We hypothesize that this is because the prior features of this dataset has low diversity and are not informative enough to lead to improved performance with data instance prior training.

**Impact of loss function**   we analyze how DIP performs when loss function is changed. We compare between the following three loss functions: hinge loss used originally in our experiments, non-saturating loss (Goodfellow et al., 2014) and the wasserstein loss (Arjovsky et al., 2017). Table 8 shows the corresponding results when DIP is used with FreezeD and DiffAugment. We observe that in case of FreezeD+DIP wasserstein loss significantly outperforms non-saturating loss and hinge loss. In case of DiffAugment hinge loss performs best followed by non-saturating loss and wasserstein loss.

**Implementation Details**   In SN-GAN architecture, while training with data instance prior, $G_{emb}$ and $D_{emb}$ are matrices which linearly transform the pre-trained features into generator conditional space of dimension 128 and discriminator feature space of dimension 1024. For baseline training,

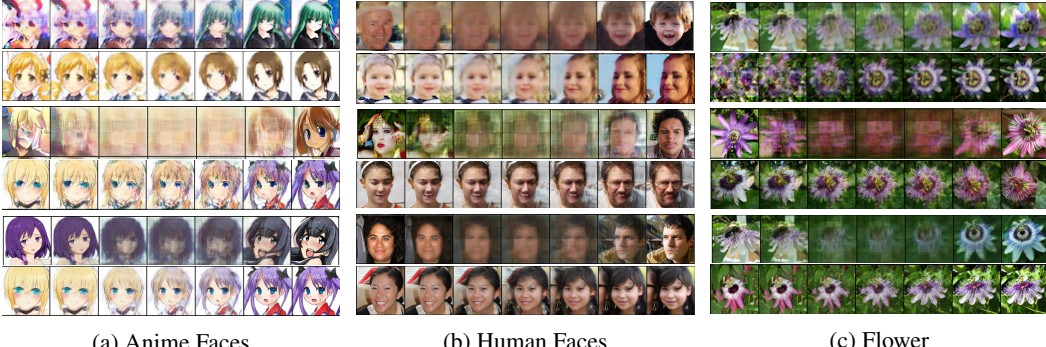

|   (a) Anime Faces   |   (b) Human Faces   |   (c) Flower   |

Figure 7: Few-shot interpolation samples between instance-level priors: Scratch *(Row 1)*, Scratch + DIP-Vgg16 *(Row 2)*, FreezeD *(Row 3)*, FreezeD + DIP-Vgg16 *(Row 4)*, DiffAugment *(Row 5)*, DiffAugment + DIP-Vgg16 *(Row 6)*

| Method | Style-GAN 2 (256 x 256) | | |
| --- | --- | --- | --- |
| | **Panda** FID ↓ | **Grumpy Cat** FID ↓ | **Obama** FID ↓ |
| FreezeD | 16.69 | **29.67** | 62.26 |
| + DIP-Vgg16 | **14.66** | 29.93 | **54.87** |
| | | | |
| DiffAugment | 12.06 | **27.08** | 46.87 |
| + DIP-Vgg16 | **11.14** | 28.45 | **43.79** |
| | | | |
| BSA* | 21.38 | 34.20 | 50.72 |
| GLANN + DIP-Vgg16 | **11.51** | **29.85** | **38.57** |

Table 6: 100-shot image generation results using StyleGAN-2 (Karras et al., 2020b) architecture on Panda, Grumpy-cat and Obama datasets. FID is computed between 5k generated and the complete training dataset. * denotes directly reported from the paper (Zhao et al., 2020b).

| Method | Anime (SNGAN) FID ↓ |
| --- | --- |
| FreezeD + DIP | **93.36** |
| FreezeD + Logo-GAN (K=5) | 226.60 |
| FreezeD + Logo-GAN (K=10) | 183.38 |
| | |
| DiffAugment + DIP | **48.67** |
| DiffAugment + Logo-GAN (K=5) | 130.54 |
| DiffAugment + Logo-GAN (K=10) | 190.59 |

Table 7: 100-shot image generation comparison of DIP with Logo-GAN (Sage et al., 2018) on Anime dataset using Vgg16 network trained on ImageNet. FID is computed between 10k generated and real samples (disjoint from training set).

we use an embedding for each of the 100 training images to ensure minimal difference between baseline and our approach without increasing number of parameters. We also experimented with self-modulated (Chen et al., 2018) and unconditional training which resulted in either training collapse or worse results in all approaches. In DiffAugment, we use three augmentations: translation, cutout, and color with consistency regularization hyperparameter as 10 and training is done from scratch following the implementation in their paper (Zhao et al., 2020b). In FreezeD, we freeze the first five blocks of the discriminator and finetune the rest. We use spectral normalization for both generator and discriminator during training with batch size of 25, number of discriminator steps as 4, $G$ and $D$ learning rate as $2e - 4$, $\mathbf{z}$ dimension as 120 and maximum number of training steps as $30K$. During evaluation, moving average weights (Salimans et al., 2016a) of the generator is used in all experiments unless stated otherwise. For FID calculation, we select the snapshot with best FID similar to (Chen et al., 2019; Zhao et al., 2020b). For calculating precision and recall based on the k-nearest neighbor graph of inception features, as in (Kynkäänniemi et al., 2019), we use $k$ as 10 for Precision and 40 for Recall.

For StyleGAN-2, $G_{emb}$ is a 2-layer MLP with ReLU non-linearity which maps $C(\mathbf{x})$ to a 512-dimensional generator conditional space. It is then concatenated with random noise $\mathbf{z}$ of dimension 512 which is used as input in the mapping network. $D_{emb}$ is a linear transformation matrix and discriminator loss is projection loss combined with real/fake loss. Training is done with batch-size of 16 for DiffAugment[3] and 8 for FreezeD[4] till $20k$ steps.

In case of BSA, we show that DIP can be used to improve the results on similar non-adversarial generative models. Specifically, we perform experiments with GLANN [5] which is a two step training

---

[3]https://github.com/mit-han-lab/data-efficient-gans

[4]https://github.com/sangwoomo/FreezeD

[5]https://github.com/yedidh/glann

| Method | Pre-training | SN-GAN (128 x 128) | | | | | |
| | | Anime | | | Faces | | |
| | | H | NS | W | H | NS | W |
| FreezeD | ✓ | 109.40 | 102.43 | 148.99 | 107.83 | 105.34 | 209.23 |
| + DIP-Vgg16 | | 93.36 | 82.49 | 74.91 | 77.09 | 77.38 | 71.05 |
| DiffAugment | ✗ | 85.16 | 106.96 | 252.11 | 109.25 | 107.18 | 325.85 |
| + DIP-Vgg16 | | 48.67 | 48.61 | 56.43 | 62.44 | 68.66 | 81.03 |

Table 8: Comparison of loss function in few-shot image generation using 100 training images (FID: lower is better). H is hinge loss, NS is non saturating loss and W is wasserstein loss.

procedure, as follows: (1) Optimize for image embeddings $\{\mathbf{e}_i\}$ of all training images $\{\mathbf{x}_i\}$ jointly with a generator network $G$ using perceptual loss; and (2) Learn a sampling function $T : \mathbf{z} \rightarrow \mathbf{e}$ through IMLE for generating random images during inference. For adding data instance prior in the training procedure of GLANN, instead of directly optimizing for $\{\mathbf{e}_i\}$, we optimize for the following modified objective:

$$\underset{G,G_{emb}}{\arg\min} \sum_i L_{perceptual}(G \circ G_{emb} \circ C(\mathbf{x}_i), \mathbf{x}_i)$$

$$\text{where } \{e_i\} = \{G_{emb} \circ C(\mathbf{x}_i)\}$$

(7)

We finetune the pre-trained generator on batch-size of 50 with a learning rate of 0.01 for 4000 epochs. During second step of IMLE optimization, we use a 3-layer MLP with $\mathbf{z}$ dimension as 64 and train for 500 epochs with a learning rate of 0.05.

**Comparison with Logo-GAN (Sage et al., 2018)** Logo-GAN has shown advantage of using features from pre-trained ImageNet network in unconditional training by assigning class label to each instance based on clustering in the feature space. We compare our approach with this method in the few-shot data setting. For implementing logo-GAN, we perform class-conditional training (Miyato et al., 2018) using labels obtained by K-means clustering on Vgg16 features of 100-shot Anime dataset. The results reported in Table 7 show the benefit of directly using features as data instance prior instead of only assigning labels based on feature clustering.

## B LIMITED DATA IMAGE GENERATION

**Experiments on CIFAR-10 and CIFAR-100** We experiment with unconditional BigGAN and StyleGAN2 model on CIFAR-10 and CIFAR-100 while varying the amount of data as done in (Zhao et al., 2020b). We compare DIP with DiffAugment on all settings and the results are shown in Table 9. In the limited data setting (5% and 10%) augmenting DiffAugment with DIP gives the best results in terms of FID for both BigGAN and StyleGAN2 architectures. When trained on complete training dataset DIP slightly outperforms DiffAugment on BigGAN architecture. BigGAN model used for training CIFAR-10 and CIFAR-100 is same as the one used for large scale experiments in Section 5.3. In DiffAugment with BigGAN architecture, we use all three augmentations: translation, cutout, and color along with consistency regularization hyperparameter as 10. In DiffAugment + DIP consistency regularization hyperparameter is changed to 1. For experiments on StyleGAN2 architecture we use the code-base of DiffAugment [6].

**Implementation details of experiment on 128 Resolution datasets in Section 5.2** We use our approach in conjunction with existing methodologies in a similar way as the few-shot setting with $G_{emb}$ and $D_{emb}$ as linear transformation matrices which transform the data priors into the generator's conditional input space of dimension 128 and discriminator feature space of dimension 1536. During baseline training, we use self-modulation (Chen et al., 2018) in the batch-norm layers similar to (Chen et al., 2019; Schonfeld et al., 2020). In DiffAugment, we use three augmentations: translation, cutout, and color with consistency regularization hyperparameter as 10. During FreezeD training, we freeze the first 4 layers of discriminator. For TransferGAN, FreezeD, MineGAN and its augmentation with DIP, we use the following hyperparameter setting: batch size 256, $G$ and $D$

---
[6]https://github.com/mit-han-lab/data-efficient-gans/tree/master/DiffAugment-stylegan2

| Method | CIFAR-10 | | | CIFAR-100 | | |
|---|---|---|---|---|---|---|
| | 100% data | 20% data | 10% data | 100% data | 20% data | 10% data |
| BigGAN | 17.22 | 31.25 | 42.59 | 20.37 | 33.25 | 42.43 |
| +DIP | 9.70 | 16.24 | 27.86 | 12.89 | 21.70 | 31.48 |
| +DiffAugment | 10.39 | 15.12 | 18.56 | 13.33 | 19.78 | 23.80 |
| +DiffAugment & DIP | **9.52** | **14.24** | **18.50** | **12.70** | **16.91** | **20.47** |
| StyleGAN2* | 11.07 | 23.08 | 36.02 | 16.54 | 32.30 | 45.87 |
| +DiffAugment* | 9.89 | 12.15 | 14.5 | 15.22 | 16.65 | 20.75 |
| +DiffAugment & DIP | **9.50** | **10.92** | **12.03** | **14.45** | **15.52** | **17.33** |

Table 9: Comparison of FID on CIFAR-10 and CIFAR-100 while varying the amount of data used during training. Above all approaches are trained with random-horizontal flip augmentation of real images. BigGAN-DiffAugment includes consistency regularization (Zhang et al., 2019) following the implementation provided by authors (Zhao et al., 2020b). Best FID values are reported for each model. * denotes directly reported from paper.

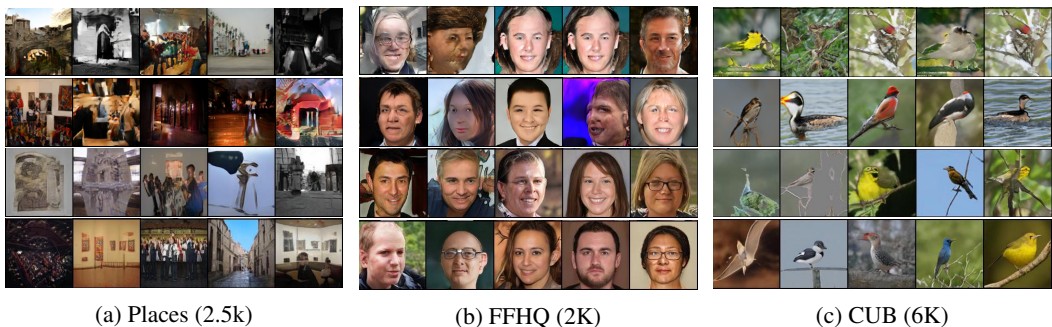

(a) Places (2.5k)  (b) FFHQ (2K)  (c) CUB (6K)

Figure 8: Sample generated images on limited data training: FreezeD *(Row 1)*, FreezeD + DIP-Vgg16 *(Row 2)*, DiffAugment *(Row 3)* and DiffAugment + DIP-Vgg16 *(Row 4)*

lr $2e-4$ and $\mathbf{z}$ dimension 120. For DiffAugment, batch size is 32, D-steps is 4 and rest of the hyperparameters are same. Training is done till 30k steps for DiffAugment, FreezeD, and 5k steps for the rest. The moving average weights of the generator are used for evaluation. We use pre-trained network from [7] (Brock et al., 2018) for finetuning.

## C  LARGE-SCALE IMAGE GENERATION

**Test for memorization in trained model**   For analyzing memorization in GANs, we evaluate it on the recently proposed test to detect data copying (Meehan et al., 2020). The test calculates whether generated samples are closer to the training set as compared to a separate test set in the inception feature space using three sample Mann-Whitney U test (Mann & Whitney, 1947). The test statistic $C_T << 0$ represents overfitting and data-copying, whereas $C_T >> 0$ represents underfitting. We average the test statistic over 5 trials and report the results in Table 10. We can see that using data instance priors during GAN training does not lead to data-copying according to the test statistic except in case of FFHQ dataset where both DIP and baseline $C_T$ values are also negative.

**Image inversion**   To invert a query image, $\mathbf{x}_q$ using our trained model, we optimize the prior (after passing it to $G_{emb}$) that is used to condition each resolution block, independently. Mathematically, we optimize the following objective:

$$\mathbf{z}^*, C_1^*, ..C_k^* = \underset{\mathbf{z}, C_1, ..C_2}{\arg\min} \|G(\mathbf{z}|C_1, ..C_k) - \mathbf{x}_q\|_2^2, \ \mathbf{x}_q^{inv} = G(\mathbf{z}^*|C_1^*, ..C_k^*)$$

Here, $C_i$ (after passing it through $G_{emb}$) is the prior that is used to condition the $i^{th} \in \{1...k\}$ resolution block. To get a faster and better convergence, we initialize all $C_i$ as $G_{emb}(C(\mathbf{x}_q))$. The optimization is achieved via back-propagation using Adam optimizer with learning rate of 0.1.

---

[7]https://github.com/ajbrock/BigGAN-PyTorch

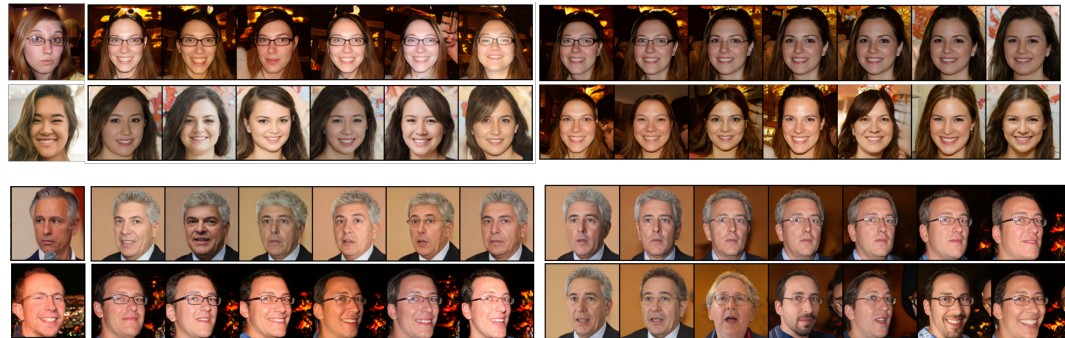

Figure 9: Semantic variations using pre-trained Vgg16 conditional DIP module on FFHQ dataset. *(Left:)* Random samples generated with prior as feature of the first image in each row; *(Right:)* first and second row in both images shows generated samples by interpolation and Bernoulli mixup between two image priors respectively.

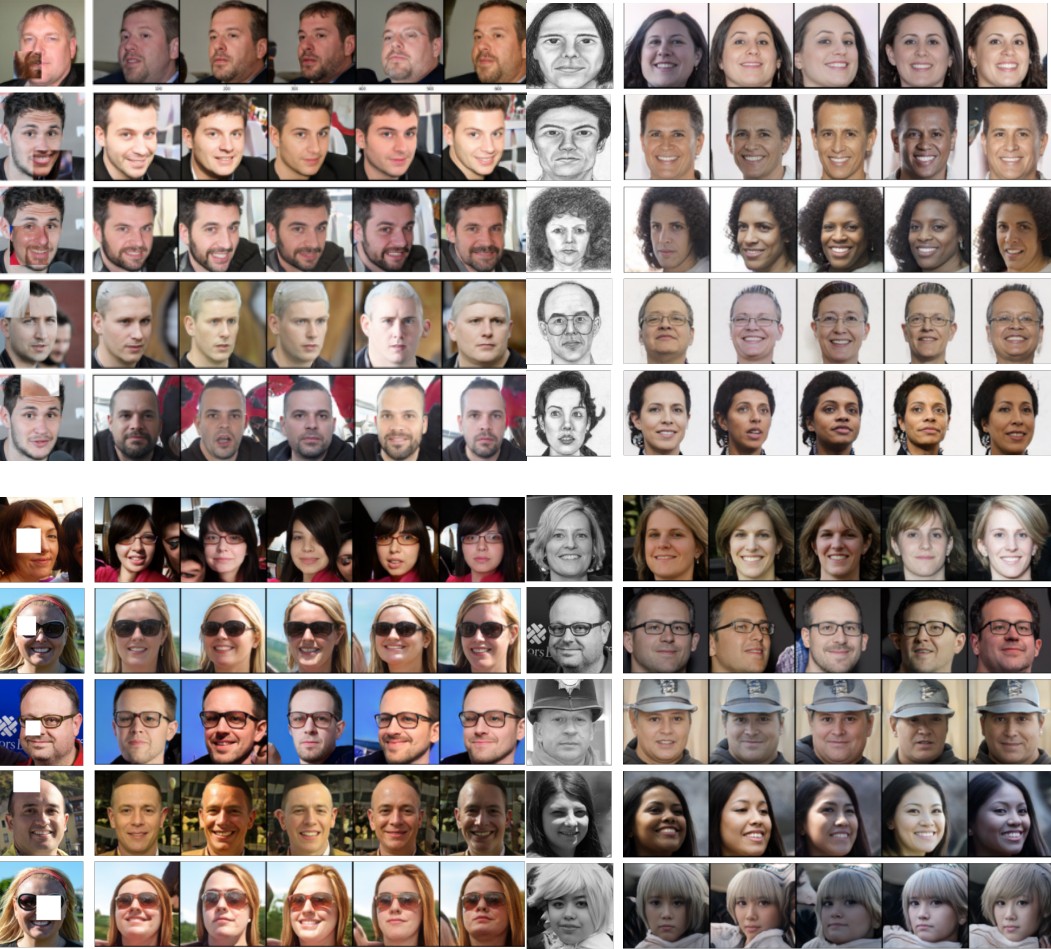

Figure 10: Examples of semantic diffusion used in image manipulation on FFHQ dataset using our DIP-Vgg16 approach. *Top-Left:* Custom Editing; *Top-Right:* Sketch-to-Image; *Bottom-Left:* Inpainting; *Bottom-Right:* Colorization

**Semantic diffusion for image manipulation**     Figure 10 shows more examples of semantic diffusion being used in standard image manipulations like colorization, editing, sketch-to-image translation and inpainting.

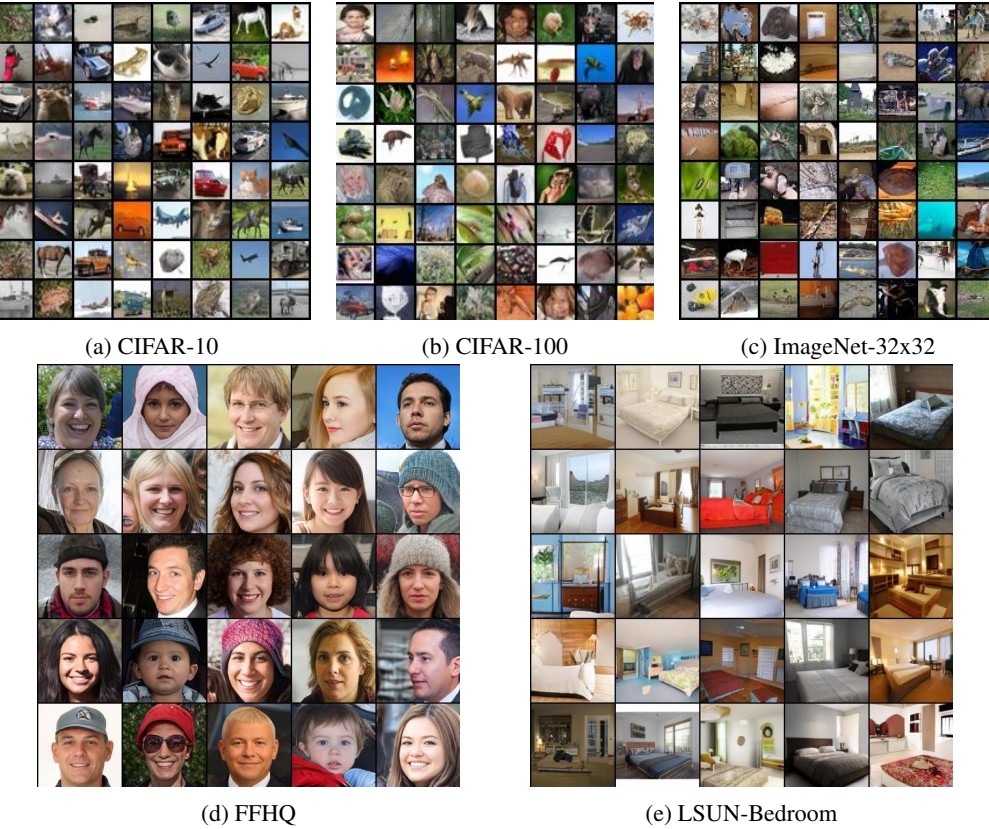

(a) CIFAR-10        (b) CIFAR-100        (c) ImageNet-32x32

(d) FFHQ              (e) LSUN-Bedroom

Figure 11: Samples generated by our DIP-Vgg16 approach on large-scale image generation

| Methods | CIFAR-10 $C_T$ | CIFAR-100 $C_T$ | FFHQ $C_T$ | LSUN $C_T$ | ImageNet32x32 $C_T$ |
|---|---|---|---|---|---|
| Baseline | 3.02 | 4.26 | -0.15 | 2.59 | 10.5 |
| DIP-Vgg16 | 1.24 | 2.50 | -0.63 | 3.31 | 7.48 |
| DIP-Vgg16 (GMM) | 1.58 | 3.05 | -0.81 | 1.06 | 8.53 |
| DIP-Vgg16 (K-means) | 1.96 | 3.70 | -0.46 | 1.12 | 8.47 |
| DIP-SimCLR | 2.23 | 3.30 | -1.14 | 2.49 | 9.70 |
| DIP-SimCLR (GMM) | 2.86 | 3.48 | -1.49 | 0.13 | 9.91 |
| DIP-SimCLR (K-means) | 2.45 | 3.91 | -1.84 | -0.12 | 10.11 |

Table 10: Test for evaluating data-copy and memorization in GANs (Meehan et al., 2020) for different approaches and datasets. Test statistic $C_T << 0$ denotes overfitting and data-copying, and $C_T >> 0$ represents under-fitting.

**Implementation Details** We use a single linear layer to transform the pre-trained image features to the generator's conditional input space of 128 dimensions, and discriminator feature space of 1024 dimensions respectively. A hierarchical latent structure similar to (Brock et al., 2018) is used during DIP training. During evaluation with K-means and GMM on ImageNet and LSUN-Bedroom we first randomly sample 200K training images and then fit the distribution since clustering on complete training set which is in the order of millions is infeasible. In the training of the unconditional baseline, we use self-modulation (Chen et al., 2018). In SSGAN, for rotation loss we use the default parameter of 0.2 for generator and 1.0 for discriminator as mentioned in (Chen et al., 2019). For training Self-Conditional GAN (Liu et al., 2020), we set the number of clusters to 100 for all datasets. For CIFAR-10 and CIFAR-100, we re-cluster at every 25k iterations with 25k samples, and for ImageNet, at every 75k iterations with 50k samples following default implementation as in (Liu et al., 2020). Following standard practice (Zhang et al., 2019), we calculate FID, Precision and Recall between test split and an equal number of generated images for-10, CIFAR-100, and ImageNet $32 \times 32$, i.e., 10k, 10k, and 50k, respectively. For FFHQ and LSUN-bedroom datasets, we use 7k and

30k generated and real (disjoint from training) samples, respectively. For all datasets and methods, training is done with batch size of $64$, G and D learning rate is set to $0.0002$, $\mathbf{z}$ dimension equals $120$ and spectral normalization is used in both generator and discriminator networks. Training is done till 100k steps for all datasets except ImageNet which is trained for 200k steps and moving average weights of generator are used during evaluation.

