# OpenReview forum: "Data Instance Prior for Transfer Learning in GANs"
_ICLR.cc/2021/Conference — Reject_

### Official Review · AnonReviewer2 · 2020-10-26
**Data Instance Prior for Transfer Learning in GANs**

**Rating:** 6
**Confidence:** 4

**Review:**

##########################################################################

Summary:

The paper focuses on  improving the performance of training generative adversarial networks (GANs) with limited target data. With the low diversity and quality when traing GANs with few data,  the paper proposes to use data instance prior to reduce the overfitting. Specially, taking the target sample as input, the data prior is extracted by a pre-trained network /  self-supervised model  , and then mapped into the embedding by  both G_emb and D_emb. The former acts as the class embedding, and the latter is the image embedding combined with the discriminator.     Authors also extend the proposed method into the large dataset, and provide the cluster method or a Gaussian Mixture Model.  The quantitative and qualitative result support the proposed method

##########################################################################

Pros:
+ The idea of utilizing the informative data prior to help train GANs is interesting.  Authors leverage the pre-trained model to extract data prior, and combine it with conditional GANs. Basically, SNGAN is considered in this peper, which uses the projection loss instead of cross-entropy loss to perform conditional image generation. Based on form of the projection loss, authors are able to avoid the problem the label of target data, and directly use the extracted embedding to conduct image generation.

+ For  large dataset, authors also provide a simple and effective method to get the semantic embedding.

+ For experiment, the paper uses current SOTA methods (BigGAN, SNGAN and StyleGAN2) and a series of datasets to evaluate the proposed method. Besides, all latest methods (to my best knowledge)  is compared to the proposed method, even the unpublished papers (DiffAugment),  which indicates that the proposed method is effective and convincing .

##########################################################################

 Cons:
- For me, the paper is so clear to understand, and miss a few information.
  (1) Does the final objective contains E.q 2 ?  From the description above E.q 3 and the architecture , it seems to contain two losses.

  (2) Is E.q 2 used at inference time which is combined to the Inference section? I fail to connect the inference section to the whole paper.

  (3) If E.q 2 is utilized in this paper, what is different to BSA?  From my point, BSA assigns the noise and the real sample pair, and optimize  the input noise as well as the parameter of the batchnorm,  but fails to consider the adversarial loss,  which results in generating blur image. In this paper, authors  additionally consider the adversarial loss, and improve the reality of the synthesized image.


- With conditional GANs is selected in this paper, I am wondering how to combine to the StyleGAN2, although the description is provided in Appendix. To be honest, I am not sure it works well when combining StyleGAN2 with project loss.

- What is the goal of both G_emb and D_emb? Do them just map the extracted embedding  to the same dimension required by conditional GANs?  authors mention that  it is   on-linearity  or   linear transformation matrices for different GANs frameworks, and  varying for  varying GANs architectures.  What is  linear transformation matrices?  what is the role to design    both G_emb and D_emb?

- In table 1, the result when combining FreezeD on Flower is low. Could authors explain it?

Minor comments:
(1) Is  TransferGAN cited correctly in section of Baselines and Datasets  in page 6 ?  I think it is the one [1], which is first paper to perform transfer learning for GANs with limited data.

(2) What is the x and hat of x in E.q 3.

(3) In abstract,  authors mention ' Previous works have addressed training inlow data setting by leveraging transfer learning and data augmentation techniques with limited success'. Is it  'with limited success' or 'with limited data'?

[1] Transferring gans:  generating images from limited data.

---

> ### Author Response · Authors · 2020-11-25
> **Response to Reviewer 2**
>
> We would like to thank the reviewer for providing valuable feedback for the paper. We are pleased that the reviewer finds our approach interesting and the experiments effective. We now clarify the raised concerns.
>
> **Q1 : Does the final objective contain E.q 2? E.q 3 seems to contain two losses?**
>
> Our final training objective is now mentioned in Eq3 that is the addition of real/fake loss with projection loss similar to c-GAN [3].
>
> **Q2 : Is E.q 2 used at inference time which is combined to the Inference section? I fail to connect the inference section to the whole paper.**
>
> Apologies for this confusion. No, Eq 2 of IMLE is not used either during training or at the inference stage of our methodology. As DIP trained models are essentially conditional GANs, during the inference stage it requires both a data instance prior $C(x)$ and $z$ as inputs to generate images ie. $G(z| C(x))$.
>
> In the case of few-shot generation, as there are a few training images (~100), we simply save the corresponding training image’s prior features and perform interpolation in this prior feature set to obtain more conditional prior for inference (Eq. 5 in paper).
> In the case of large-scale training, storing features of the complete training set becomes memory inefficient, therefore to avoid saving the large set of training priors, we propose to learn a distribution over the training image priors (using clustering or GMM)  to enable sampling of priors (Eq 6 in the paper).
> Based on the above concerns, we have rewritten our Methodology section to incorporate these comments by updating the Eq3 with the final loss of real/fake loss with projection loss and adding a pseudo-code of our DIP training.
>
> **Q3: If E.q 2 is utilized in this paper, what is different to BSA?**
>
> We don’t use E.q 2 in our paper. We mention IMLE since it serves as motivation for our approach of having an instance level condition that promotes images generated using prior $C(x)$ of image $x$ (ie. $G(z|C(x))$ ) to be close to $x$ in discriminator space but the specific formulation as described in Eq 2 is not used in our methodology.
>
> **Q4: How well does StyleGAN2 work with projection loss?**
>
> In comparison with BigGAN, we observe that performance improvement of projection loss is less in StyleGAN2 architecture in the case of few-shot data settings.
> Therefore, we have also added results on using our conditional architecture of styleGAN2 as described in Appendix for CIFAR10 and CIFAR100 datasets which shows improvement over DiffAug[2] method in limited data setting:
>
> | Method || CIFAR-10 ||| CIFAR100 ||
> |-||-|||-||
> |   |100% | 20% | 10% | 100% | 20% | 10%|
> |StyleGAN2 DiffAug | 9.89 | 12.15 |14.5 |15.22 | 16.65 | 20.75 |
> |StyleGAN2 DiffAug+DIP | 9.50 | 10.92 | 12.03 | 14.45 | 15.52 | 17.33 |
>
>
> We note that the Authors of ADA [1] have also shared results on using projection loss in StyleGAN2 architecture for CIFAR10 for class conditional image generation.
>
> **Q5: What is the role of design and hence the goal of both G_emb and D_emb? Do they just map the extracted embedding to the same dimension required by conditional GANs? What are linear transformation matrices?**
>
> Yes, the goal of both $G_{emb}$ and $D_{emb}$ is to map the extracted embedding to the dimension required by conditional GAN as used in BigGAN architecture.
>
> G_emb maps the extracted prior to a shared embedding space which is then used as input to all conditional batch-norm layers in the generator. It reduces the dimension of shared embedding space (usually 128) as compared to the dimension of the extracted pre-trained feature prior (512 for Vgg-16 and 2048 for SimCLR).
> D_emb maps the extracted prior to the same dimension as discriminator features to apply projection loss similar to class embedding matrix in cGANs[3].

---

> > ### Author Response · Authors · 2020-11-25
> > **Response to Reviewer 2 continuation**
> >
> > **Q6: In table 1, the result when combining FreezeD on Flower is low. Could the authors explain it?**
> >
> > For experiments on the Flowers dataset (that is a subset of Flowers dataset containing only the "passion" class of flower), DIP performs only slightly better (TransferGAN, DiffAugment, and BSA) or worse (for e.g. in FreezeD by FID metric). Here, FID is calculated using only 251 real images from the reference distribution as compared to 10k/7k separate test-set available for the Anime and Faces datasets respectively. We believe that FID calculated using less number of real images makes it to be a less reliable indicator of the generator’s performance [4].
> > We also observe that just by memorizing given 100 training images it is possible to achieve an FID of 66.91. On analyzing the generated images by the baseline model, we observe that it overfits to given 100 training images with poor interpolation between conditional embeddings as shown in Fig 6 in the paper. We believe that because of this overfitting, the FID metric of the baseline is better when the sample size of real images is small. In the table below, we show the FID of the baseline and DIP model.
> >
> > |   | Baseline | DIP  | Train set |
> > |-|-|-|-|
> > | FID  |91.80 | 120.43 | 66.91 |
> >
> > To further analyze this, we conduct an ablation experiment, where we create another dataset of Flowers which we call Flower-Diverse. This dataset is created by randomly sampling one image from 100 out of 102 classes of oxford flowers dataset, hence creating a 100 training image dataset. Remaining images from these 100 classes (~8000 images) are used as images from real distribution in FID calculation. Here, we observe the benefit of DIP with baseline approaches as shown below on the Flower-Diverse dataset.
> >
> > |Flower-Diverse | FreezeD |DiffAugment |
> > |-|-|-|
> > | Baseline | 90.01 | 82.01 |
> > | +DIP-Vgg16 | 86.55 | 50.56 |
> >
> > **Minor comments:**
> >
> > (1) Is TransferGAN cited correctly in the section of Baselines and Datasets in page 6 ?
> >
> >  We thank the reviewer for pointing this out.
> >
> > (2) What is the x and hat of x in E.q 3.
> >
> > As in GANs, each training step comprises of $d_{steps}$ of Discriminator update and a single step of Generator update, a popular choice for $D_{step}$ is 4, and the same is used for our BigGAN experiments. We denote $x$ and $\tilde{x}$ as images sampled from the real distribution for $G$ and $D$ updates respectively. For our implementation purpose where $D_{steps} = 4$ and $G_{step} = 1$, we take x same as the last batch (i.e 4th batch in case when $D_{steps}=4$) of sampled images from true distribution for $D$ update, thus $x$ is a subset of $\tilde{x}$. We have now removed this notation from our paper since the updated Eq 3 now mentions loss for $G$ and $D$ separately and have added a pseudo code in the methodology section for clarity.
> >
> > (3) In the abstract, Is it 'with limited success' or 'with limited data'?
> >
> >  It’s with limited data. We have rectified this in the paper.
> >
> > References:
> >
> > [1] Karras T, Aittala M, Hellsten J, et al. Training generative adversarial networks with limited data[J]. arXiv preprint arXiv:2006.06676, 2020.
> >
> > [2] Zhao S, Liu Z, Lin J, et al. Differentiable augmentation for data-efficient gan training[J]. arXiv preprint arXiv:2006.10738, 2020.
> >
> > [3] Takeru Miyato and Masanori Koyama. cgans with projection discriminator. In International Conference on Learning Representations, 2018
> >
> > [4] Esther Robb et al. Few-Shot Adaptation of Generative Adversarial Networks, arXiv preprint arXiv:2010.11943, 2020

---

### Official Review · AnonReviewer4 · 2020-10-28
**GAN with Transfer Learning**

**Rating:** 7
**Confidence:** 4

**Review:**

This paper illustrates how they train GANs with small sample sizes with the help of Transfer Learning. The paper tackled a very specific problem: what should we do with a small sample training size if we want to train a GAN. The authors have supported their arguments by a proof in Data In Prior and experiment results. They illustrated well in both aspects.

Here are my point of views:
Transfer learning is a good way to help GANs when sample size is limited. but I have two concerns over this paper:
1. The datasets are very popular in the filed of GAN, however, for the Anime one, I am just curious how the VGG pretrained network can also help.

2.  As for the experiment, it lacks of comparison with results that transfer learning is not applied.

Generally, the paper is good and it can help data augmentation in other applications.

---

> ### Author Response · Authors · 2020-11-25
> **Response to Reviewer 4**
>
> We would like to thank the reviewer for providing valuable feedback for the paper. We are pleased to know that the reviewer finds our experiments and DIP’s application effective. We now answer the raised concerns.
>
> **Q1: How can VGG pretrained network also help for Anime dataset ?**
>
> To examine the usefulness of Vgg features on Anime dataset, we evaluate it on the anime character classification task. We took a subset of 70k images from the Anime Face dataset that had labels assigned among the 50 character tags. Each character tag has around 1000-1500 images. We train a single linear classifier on VGG-16 features of 50k samples and evaluate it on the rest 20k samples. We observe an accuracy of ~75% and ~67% on training and test sets respectively. When a single linear classifier is trained upon SimCLR features, the respective accuracies were ~81% and ~63.5%. This highlights that even for fine-grained and out of domain distributions like Anime, pretrained VGG-16 features are semantically rich enough to achieve a decent classification score.
>
>
> **Q2: As for the experiments, it lacks a comparison with results that transfer learning is not applied.**
>
> We have now added the results of baseline training when weights of the network are randomly initialized (denoted as Scratch) and Scratch with DIP in the few-shot setting in Table 1. We observe that FID scores are better than TransferGAN+DIP and interpolation between conditional prior is shown in Fig 7.
>
> | | Anime | Faces | Flower |
> |-|-|-|-|
> |Scratch | 120.38 | 140.66 |124.02 |
> |+DIP |  66.85 | 68.49 | 94.22 |
>
>
> We also show comparison of DIP with baseline without transfer learning (i.e random initialization of model weights) for limited data setting on CIFAR-100 with BigGAN architecture while varying the dataset size. We have added these results in the Appendix Table 9.
>
> | CIFAR-100 | 100% | 20% | 10% |
> |-|-|-|-|
> |BigGAN Baseline | 20.37 | 33.25 |42.43 |
> |BigGAN +DIP | 12.28 | 21.70 | 31.48 |

---

### Official Review · AnonReviewer3 · 2020-10-28

**Rating:** 6
**Confidence:** 4

**Review:**

This submission deals with transfer learning for training GANs with limited label data. The challenge is that training with limited data can result in mode collapse. This submission proposes to use data priors for each instance of the target distribution, transformed through knowledge from a source domain, as conditional information in GAN to ensure mode coverage of the target data distribution. A pre-trained feature extractor is used to provide the information to condition the GAN. A range of experiments is performed with the features extracted from VGG16, SIMCLR. They show consistent improvements in the image quality and diversity, measured via FID and precision-recall,  for few-shot, limited data, and even large scale data settings.

Strong points
- The idea of creating useful guides based on pre-trained features for conditional GANs makes sense.
The experiments are extensive and they show improved quality and diversity for a wide range of settings.

Correctness
- The idea and the reported experiments make sense.

Reproducibility
- The code has been provided with the code for pre-training and other details included.

More suggestions and comments
- IMLE description in (2) is vague. It seems that a gaussian sample z is taken, but on the other hand, for a given network G, z is optimized to match x. The restrictions on z is unclear.
It seems that hinge-loss is used for GAN based on (1)? It would be important if the authors could comment on the choice of the divergence/loss in this setting. One may wonder that the limited data and mode-collapse could be better handled with Wasserstein distance. An ablation study would be very useful to clarify the role of the distance.

---

> ### Author Response · Authors · 2020-11-25
> **Response to Reviewer 3**
>
> We would like to thank the reviewer for providing constructive feedback for the paper.  We are pleased to know that the reviewer finds our DIP approach interesting and our experiments extensive in the paper. We now address the raised concerns and provide results for the suggested experiments.
>
> **Q1: IMLE description in (2) is vague. It seems that a gaussian sample $z$ is taken, but on the other hand, for a given network $G$, $z$ is optimized to match $x$. The restrictions on $z$ are unclear.**
>
> We have re-written the IMLE description and its corresponding Eq. 2 to add the restrictions on $z$ in the paper and we hope that this will increase its clarity. In IMLE, $z$ is sampled from a normal distribution, and during each update generator is optimized to reduce the distance between real sample x and its nearest neighbor $G(z)$ from that mini-batch of sampled $z$’s.
> Eq 2 of IMLE is not used either during training or the inference stage for our methodology. Our final training objective is now mentioned in Eq3 that is the addition of real/fake loss with projection loss similar to c-GAN [1].
> Also, we have updated our Methodology section by modifying the Eq3 with the final loss of real/fake loss with projection loss and added a pseudo-code of our DIP training.
>
> **Q2: It seems that hinge-loss is used for GAN based on (1)? It would be important if the authors could comment on the choice of the divergence/loss in this setting. One may wonder that the limited data and mode-collapse could be better handled with Wasserstein distance.**
>
> As suggested by the reviewer, we also ran an experiment to analyze the role of loss/divergence used during GAN training. The table below shows the FID score of models trained using (1) Non-saturating loss (NS) (2) Wasserstein loss (W) and (3) Hinge loss (H) (used in the paper).
>
> | Method || Anime ||| Faces ||
> |-||-|||-||
> | | NS | W | H | NS | W |H |
> |FreezeD | 102.43 |  148.99  | 109.40 | 105.34 | 209.23 | 107.83 |
> | FreezeD + DIP-Vgg16 | 82.49 | 74.91 | 93.36 | 73.38 | 71.05 | 77.09 |
> |DiffAug |106.96 | 252.11 | 85.16 | 107.18 | 325.85 | 109.25 |
> |DiffAug + DIP-Vgg16 |48.61 | 56.43 | 48.67 | 68.66 |  81.03 | 62.44 |
>
> We use gradient penalty with Wasserstein loss in FreezeD but not in DiffAugment as it already consists of consistency regularization loss and using both leads to unstable training. Wasserstein loss works significantly better in the case of FreezeD+DIP but worse when used with the DiffAugment training strategy.  We have added this ablation experiment’s result in the Appendix section table 8 in our paper.
>
> Reference:
>
> [1] Takeru Miyato and Masanori Koyama. cgans with projection discriminator. In International Conference on Learning Representations (ICLR), 2018

---

### Official Review · AnonReviewer1 · 2020-10-29
**A transfer learning method for GANs in the limited data domain**

**Rating:** 4
**Confidence:** 5

**Review:**

This work proposed a transfer learning method for GANs in the limited data domain. It borrow ideas from IMLE (to overcome mode-collapse) and conditional GAN (to improve training stability and generation quality), by introducing data instance prior (plays a similar role to that of  label information in conditional GAN) and knowledge distillation techniques, the model is claimed to be effective in preventing mode collapse and discriminator overfitting.

Though the main idea makes sense to some extent, the writing is a little weak, especially the equations and some statements that are not correctly verified in the experiments, making it difficult to go through the paper.

Detailed concerns are listed below.

1. Eq 3 is not correct, the sample process should be placed under “Expectation” rather than “minimize”; also, there is no information about which parameter is going to be optimized here. What’s the relation ship between x and x~, are they independently sampled from the target dataset?

2. Figure 1 is confusing. It shows that the adversarial loss is depend on the projection loss, but there is no equation to show the relationship between these two losses.

3. There are many network components in the proposed method, it is not easy to guess the objective related to the real/fake score in an adversarial manner, a clear equation for the adversarial training is necessary. Also, a detailed training process is necessary, e.g. how to sample, when to optimize the generator, discriminator, and the embedding networks?

4. How to make sure that “enforcing feature Df(G(z|C(x))) to be similar to Demb(C(x)) ensures that for each real sample, there exists”? Apparently, the constrain is made on some latent space and there is no statement to show that close in latent space is equivalent to close in image space.

5. When the data is large scale, doing clustering is inhibitive.

6. On Table 3, the results are not state of the art on CIFAR10 and fall behand the recent works on generation with limited data, e.g. discriminator augmentations (DA) [1],  DiffAugment [2], then what’s the advantage of the proposed method when compare to these methods?

7. An ablation study is suggested. It’s not easy to see the contribution of each part (e.g. knowledge distillation, covering real data modes) to the final performance.

8. It claims that the proposed method can prevent discriminator overfitting, but in the experiments, it is not shown.

9. What’s the motivation to do interpolation on the data instance prior? How to make sure that the interpolation on the prior (Eq.4) is smooth?

10. On semantic diffusion for image manipulation. It is not clear how to obtain the results shown in Figure 4/9 on custom editing, e.g. there is only one input image, how to compute and exchange the C(x)? Also, the effects of manipulating high-level semantics and fine-grained details are not observed/discussed in the experiments.


[1] Karras T, Aittala M, Hellsten J, et al. Training generative adversarial networks with limited data[J]. arXiv preprint arXiv:2006.06676, 2020.
[2] Zhao S, Liu Z, Lin J, et al. Differentiable augmentation for data-efficient gan training[J]. arXiv preprint arXiv:2006.10738, 2020.

---

> ### Author Response · Authors · 2020-11-25
> **Response to Reviewer 1 (Question 1-4)**
>
> We would like to thank the reviewer for providing insightful ideas and constructive comments on the paper.
> At the outset, we wish to point out that we have comprehensively addressed (rewritten) all the concerns regarding clarity in our methodology (which seemed to be the major concern across the reviews). We have also introduced an algorithm to show the implementation details of our method clearly. We now address each of the concerns pointed out by the reviewer.
>
> **Q1: The sampling process should be placed under “Expectation” rather than “minimize” in Eq3. What parameters are getting optimized in the loss objective in Eq 3? What is the relationship between $x$ and $\tilde{x}$ ?**
>
> Yes, we thank the reviewer for pointing out the correct position of the sampling process.  We have updated Eq. 3 in our paper to reflect the objective being optimized for both $G$ and $D$ parameters. The final discriminator output which is optimized adversarially for $G$ and $D$ is the addition of real/fake loss with projection loss between discriminator feature and conditional prior embedding. We have also added a pseudo-code in our Methodology section to further clarify our training process of DIP.
> Generally, each training step of GAN comprises of $d_{steps}$ of Discriminator update and a single step of Generator update. We denote $x$ and $\tilde{x}$ as images sampled from the real distribution for $G$ and $D$ updates respectively. For our implementation purpose where $d_{steps} = 4$ and $G_{step} = 1$, we take $x$ as the last batch (i.e 4th minibatch) of sampled real images for $D$ update, thus making $x$ a subset of $\tilde{x}$. We have now removed this notation from our paper since the updated Eq 3 now mentions loss for $G$ and $D$ separately.
>
>
> **Q2: Figure 1 is confusing, it shows that the adversarial loss is dependent on the projection loss. What is the equation that shows the relationship between these two losses?**
>
> Yes, the adversarial loss depends on the projection loss. The final loss for our DIP approach is the addition of Real/Fake loss and projection loss similar to the cGAN objective [4] that is adversarially optimized. We have modified Figure 1 in our paper to remove this confusion and have updated Eq 3 that reflects this.
>
> **Q3: What is the final equation for the adversarial training for DIP? Also, a detailed training process is necessary, e.g. how to sample, when to optimize the generator, discriminator, and the embedding networks?**
>
> Apologies for this confusion. We have updated Eq 3 to contain the full objective loss with respect to each component of GAN, and have now included the pseudo-code (algorithm) of DIP in our Methodology section 4 as Algorithm 1 that describes our training process in detail.
>
> **Q4: How to make sure that “enforcing feature Df(G(z|C(x))) to be similar to Demb(C(x)) ensures that for each real sample, there exists” as the constraint is made on latent space and not in the image space?**
>
> During training using DIP, images generated with condition $C(x)$ are enforced to have it’s discriminator feature close to $D_{emb}(C(x))$. Fig 9 in the Appendix qualitatively shows that for a model trained using DIP, the image generated using $C(x)$ i.e. $G(z|C(x))$ is semantically closer to the image $x$. We have also quantitatively measured this as performance on the Recall and IvoM metric (Table 4), and observe that it outperforms the baseline methods.
>
> To validate the concern regarding the equivalence of closeness in latent and image space, we measure the correlation between cosine similarity in Discriminator feature ( $D_f(.)$ ) and Vgg-16 feature (perceptual similarity) space. Vgg-perceptual similarity [3] is an accepted measure of image similarity and has been used in generative models like IMLE, GLANN, BSA as a proxy for constraints in image space. Additionally, we also report the correlation between cosine similarity in Discriminator feature space and $L_2$ closeness measure in the image space. The table below reports our findings where we observe a high positive correlation between cosine similarity in $D_f$ and VGG perceptual similarity; and a moderate negative correlation between cosine similarity in $D_f$ and $L_2$ distance in Image space.
>
> | Pearson Correlation | Anime | FFHQ | CIFAR-10 |
> |-|-|-|-|
> | $D_f$ cosine vs VGG Perceptual  | 0.65 | 0.81 | 0.80 |
> | $D_f$ cosine vs Image $L_2$ | -0.46 |-0.61 | -0.54 |
>
> To quantitatively verify that G(z|C(x)) is close to x in the trained model, we also show below the perceptual similarity between the two as compared to a random pair of images.
>
> | Cosine Similarity | Image $x$ and its conditional  generated image $G(z\|C(x))$ | Random Pair |
> |-|-|-|
> | VGG perceptual space| 0.512 $\pm$ 0.067 | 0.382 $\pm$ 0.050  |
> |Discriminator’s feature space| 0.59 $\pm$ 0.096 | 0.50 $\pm$ 0.070  |

---

> > ### Author Response · Authors · 2020-11-25
> > **Response to Reviewer 1 (Question 5-7)**
> >
> > **Q5: When the data is large-scale, doing clustering becomes inhibitive.**
> >
> > Yes, we agree that clustering is inhibitive for large-scale datasets e.g. ImageNet and LSUN-Bedroom where the training data is in the order of millions. We observed this during our experiments and therefore we have reported the performance of K-means/GMM using a subset of randomly sampled 200K instances for these datasets in our paper. We have added this detail in the paper now.
> > Below we show the relationship between the number of random samples used for fitting GMM/K-means and the corresponding FID (average of 3 runs with a standard deviation of less than 1%) on the LSUN-Bedroom dataset for DIP-Vgg16 trained model on LSUN-Bedroom.
> >
> > | | 50k | 100k | 200k | 500k | $D_{prior}$ (3 M) |
> > |-|-|-|-|-| - |
> > | GMM | 4.99 | 4.92 |4.81 | 4.43 |3.77 |
> > |Time (in secs) | 383.96 | 1063.99 | 1993.93 | 4397.56 | - |
> > | K-means | 3.84 | 4.20 |4.72 | 5.36 | - |
> > | Time (in secs) | 210.67 | 546.57 | 1344.08 | 7072.34 | - |
> >
> > This experiment was performed on a system with 32 CPU cores, 64 GB RAM, and processor Intel(R) Xeon(R) CPU @ 2.20GHz. We will be happy to include these details in the Appendix for clarity.
> >
> > **Q6: In Table 3, the results are not SOTA on CIFAR10 and fall behind the recent works on generation with limited data, e.g. DA [1], DiffAug [2]. What is the advantage of DIP when compared to these methods?**
> >
> > Authors of [1,2] use random horizontal flip augmentation for training. In our paper, we have reported FID scores for experiments that do not use horizontal flip augmentation.  We observe that we can achieve an unsupervised (i.e not using the class labels) FID score of $\\textbf{9.70}$ and $\\textbf{12.89}$ on CIFAR-10 and CIFAR-100 respectively on BigGAN architecture by utilizing this augmentation and keeping all other hyper-parameters the same as in Table 3. This is better than the FID score of 9.89 and 15.52 as reported in [2] on StyleGan2 architecture for CIFAR-10 and CIFAR-100 respectively. In the following table, we also show the benefit of our approach when used with DiffAug [2] technique on stylegan2 architecture. We have added these results in Table 9 of the Appendix.
> >
> > | Method || CIFAR-10 ||| CIFAR100 ||
> > |-||-|||-||
> > |   |100% | 20% | 10% | 100% | 20% | 10%|
> > |StyleGAN2 DiffAug | 9.89 | 12.15 |14.5 |15.22 | 16.65 | 20.75 |
> > |StyleGAN2 DiffAug+DIP | 9.50 | 10.92 | 12.03 | 14.45 | 15.52 | 17.33 |
> >
> > We also run DiffAug on unconditional BigGAN architecture for comparison with DIP. Below, we show improvement of using DIP in conjunction with DiffAug [2] on CIFAR-100, while varying the amount of data used in training on BigGAN architecture with random-horizontal flip augmentation and other hyperparameters same as in Table 3 of our paper.
> >
> > | CIFAR-100 | 100% | 20% | 10% |
> > |-|-|-|-|
> > |BigGAN DiffAug | 13.33 | 19.78 | 23.80|
> > |DiffAug+DIP | 12.70 | 16.91 |20.47|
> >
> > We have shown that our methodology DIP can be effectively combined with DiffAugment in limited /few-shot data settings for improved performance as shown in Tables 1 and 2.
> >
> > **Q7: An ablation study is suggested to highlight the contribution of each part (e.g. knowledge distillation, covering real data modes) in the final performance of DIP.**
> >
> > During our DIP training methodology in the few-shot setting, knowledge transfer can be done as
> >
> > (a) Initialization of GAN’s weight from a pre-trained GAN
> >
> > (b) Using conditional data priors for images extracted from a pre-trained network like Vgg-16.
> >
> > The contribution of (a) is shown in Table 1, where we have added results when the network weights are initialized from scratch (randomly initialized) vs pre-trained GAN weights as initialization for DIP training.
> >
> > To analyze the contribution of (b), we compare it to the two versions of the baseline approach.
> > Firstly, baseline-unconditional which is an unconditional GAN (replacing conditional batch-norm with standard batch-norm in SN-GAN)  and secondly a modification of baseline training named baseline-Embedding where we learn embedding (i.e. initialized from scratch) for each image during training similar to DIP approach (but in Baseline-Embedding the priors are learned rather than being derived/distilled from some pre-trained network as done in DIP). This shows the isolated advantage of knowledge distillation as priors in GAN training. Below we show the table of FID scores for comparison with DiffAugment as the baseline training strategy for 100 shot image generation on Anime, Faces, and Flower datasets.
> >
> > | Method | Anime | Faces | Flower |
> > |-|-|-|-|
> > |DiffAug-unconditional |160.18 |  154.30 | 136.32 |
> > | DiffAug-Embedding | 85.16 | 109.25 | 83.45  |
> > | DiffAug + DIP-Vgg16 | 48.67  | 62.44  | 79.86 |
> >
> > We note that all the baseline FID scores reported in Table 1 correspond to Baseline-Embedding approach as mentioned in Appendix since it performs better than Baseline-unconditional approach for our experiments in terms of the FID score.

---

> > > ### Author Response · Authors · 2020-11-25
> > > **Response to Reviewer 1 (Question 8-10)**
> > >
> > > **Q8: Show via experiments that the proposed method DIP can prevent discriminator overfitting.**
> > >
> > > We have included a similar analysis for discriminator overfitting as done in [1,2] in our Methodology section as Fig 2. We train a baseline and DIP model on 10% of CIFAR-100 dataset with BigGAN architecture. We report the real/fake discriminator score on the training/validation/generated samples during the course of training. In baseline training, the discriminator score of real images keeps on increasing while the discriminator score on validation images and FID quickly degrades. This suggests overfitting in training. However, in the case of Baseline+DIP training, the discriminator score on training and validation images remains similar and higher than the generated data’s discriminator score. Also, the FID value keeps on decreasing and saturates instead of abruptly increasing as the training progresses.
> > >
> > > **Q9: What’s the motivation to do interpolation on the data instance prior? How to make sure that the interpolation on the prior (Eq.4) is smooth?**
> > >
> > > As we observed in our experiments that the prior $C(x)$ controls the high-level details and the latent code $z$ influence the fine-grained details of images in $G(z|C(x))$. In the few-shot setting, we only have access to few data priors (~100). To generate more priors as to generate images with high diversity, we leverage the interpolation of priors.
> > > We observe smooth interpolation in the trained model between data priors without enforcing any explicit constraint as shown in Fig. 3. This shows the generalization of our DIP trained model in the prior space.
> > >
> > > **Q10: On semantic diffusion for image manipulation. It is not clear how to obtain the results shown in Figure 4/9 on custom editing, e.g. there is only one input image, how to compute and exchange the $C(x)$? Also, the effects of manipulating high-level semantics and fine-grained details are not observed/discussed in the experiments.**
> > >
> > > To perform semantic diffusion image x is manipulated (cut-mix) or picked from out of domain (sketches) and its pre-trained feature $C(x)$ from Vgg-16/SimCLR network is directly used as prior condition in GAN.
> > > For example in cutmix, given Image $I_1$ and Image$I_2$, we generate a new image that is a cutmix version of $I_1$ and $I_2$. Let the modified image be $I_x$ = cutmix($I_1$, $I_2$) which is shown as the first image in Fig5/10. To generate images similar to $I_x$ or perform semantic diffusion, we calculate the prior representation of this modified image as $C(I_x)$. Then, we use this prior as input to generate images $G(z| C(I_x))$ where z is randomly sampled from the standard normal distribution. In Fig 5, the first column corresponds to cutmix images $I_x$ and the 2nd and 3rd columns represent images generated via $G$ using the prior $C(I_x)$.
> > > We have shown results in Fig 9 where changing $z$ and keeping the prior fixed leads to change in fine-grained details of the generated image and interpolation in prior leads to high-level semantic changes.
> > >
> > > References:
> > >
> > > [1] Karras T, Aittala M, Hellsten J, et al. Training generative adversarial networks with limited data[J]. arXiv preprint arXiv:2006.06676, 2020.
> > >
> > > [2] Zhao S, Liu Z, Lin J, et al. Differentiable augmentation for data-efficient gan training[J]. arXiv preprint arXiv:2006.10738, 2020.
> > >
> > > [3] Richard Zhang, Phillip Isola, Alexei A Efros, Eli Shechtman, and Oliver Wang. The unreasonable effectiveness of deep features as a perceptual metric. In CVPR, 2018.
> > >
> > > [4] Takeru Miyato and Masanori Koyama. cgans with projection discriminator. InInternational Conference on Learning Representations, 2018

---

### Author Response · Authors · 2020-11-25
**Common Response to Reviewers/AC**

We thank all reviewers for insightful and constructive feedback. We are encouraged to note that reviewers found the approach of Data Instance Prior (DIP) as interesting/convincing (R2,R3,R4); extensive/effective quantitative and qualitative results (R2,R3,R4) and the approach makes sense (R1,R3). We have corrected all the typos and addressed any lack of clarity in the paper’s final version.
Below we provide an overview of the major updates made in the paper:


* Rewritten methodology section for increased clarity of training and inference stages. We have also added a pseudo-code for our algorithm DIP.

* Added a comparison table that shows improvement with styleGAN2 and BigGAN architecture over DiffAugment technique by varying the amount of training dataset (100%, 20%, 10%) on CIFAR10 and CIFAR100 datasets in Appendix Table 9.

* Added an ablation experiment by varying the choice of the loss (hinge, non-saturating, and Wasserstein loss) for GAN training as shown in Appendix Table 8.

* Updated Table 1 by comparing the results when transfer learning (initialization with pre-trained model weights) is not applied.

* Added a figure (as Figure 2) for comparison with DIP that suggests discriminator overfitting in the baseline approach when training is done in limited data setting.

---

### Decision · Program_Chairs · 2021-01-07
**Final Decision**

**Decision:**

Reject

**Comment:**

The paper proposes to use a feature extractor (encoder) $C(x)$, pre-trained with label supervision or contrastive learning on a large image dataset, to both regularize the discriminator's last feature layer $D_f(x)$ and encode the data $x$ itself as the conditional input of the generator $G(z|G_{emb}(C(x)))$. The main purpose is to help the training of GANs when there is a limited number of images in the target domain. A clear concern of this approach is that to generate a fake image, one will need to first sample a true image, making the model unattractive if the training dataset size is large (need to store the whole training dataset even after training). To mitigate this issue, the authors propose to fit up to 200k randomly sampled $G_{emb}(C(x))$ with a GMM with 1k components. To validate the practice of requiring a GMM (a shallow generative model) to help a GAN (a deep generative model) to generate, the authors have done a rich set of experiments under state-of-the-art GAN architectures or training methods (SNGAN, BigGAN, StyleGAN2, DiffAugment) to illustrate the efficacy of the proposed data instance prior and its compatibility with the state-of-the-art methods in a variety of settings. In the AC's opinion, the paper is missing references to 1) related work that combines VAE (or some other type of auto-encoder) and GAN, which often helps stabilize the GAN training [1,2,3], 2) VAE with a VampPrior [4], and 3) more broadly speaking, empirical Bayes related methods where the prior model is learned from the observed data (see [5] and the references therein). The potential advantages of using a VAE rather than a GMM to help a GAN to generate include: 1) there is no need to store 1k GMM components, which may require a large amount of memory; 2) there is no need to subsample the training set; and 3) the VAE and GAN can be jointly trained. The AC recommend the authors to discuss the connections to these related work in their future submission.

[1] Larsen, Anders Boesen Lindbo, et al. "Autoencoding beyond pixels using a learned similarity metric." International conference on machine learning. PMLR, 2016.

[2] Zhang, Hao, et al. "Variational Hetero-Encoder Randomized GANs for Joint Image-Text Modeling." International Conference on Learning Representations. 2019.

[3] Tran, Ngoc-Trung, Tuan-Anh Bui, and Ngai-Man Cheung. "Dist-gan: An improved gan using distance constraints." Proceedings of the European Conference on Computer Vision (ECCV). 2018.

[4] Tomczak, Jakub, and Max Welling. "VAE with a VampPrior." International Conference on Artificial Intelligence and Statistics. PMLR, 2018.

[5] Pang, Bo, Tian Han, Erik Nijkamp, Song-Chun Zhu, and Ying Nian Wu. "Learning Latent Space Energy-Based Prior Model." Advances in Neural Information Processing Systems 33 (2020).